# Adult chondrogenesis and spontaneous cartilage repair in the skate, *Leucoraja erinacea*

**Aleksandra Marconi[1], Amy Hancock-Ronemus[2,3], J Andrew Gillis[1,3]***

[1]Department of Zoology, University of Cambridge, Cambridge, United Kingdom; [2]Charles River Laboratories, Wilmington, Massachusetts, United States; [3]Marine Biological Laboratory, Woods Hole, Massachusetts, United States

**Abstract** Mammalian articular cartilage is an avascular tissue with poor capacity for spontaneous repair. Here, we show that embryonic development of cartilage in the skate (*Leucoraja erinacea*) mirrors that of mammals, with developing chondrocytes co-expressing genes encoding the transcription factors Sox5, Sox6 and Sox9. However, in skate, transcriptional features of developing cartilage persist into adulthood, both in peripheral chondrocytes and in cells of the fibrous perichondrium that ensheaths the skeleton. Using pulse-chase label retention experiments and multiplexed in situ hybridization, we identify a population of cycling *Sox5/6/9+* perichondral progenitor cells that generate new cartilage during adult growth, and we show that persistence of chondrogenesis in adult skates correlates with ability to spontaneously repair cartilage injuries. Skates therefore offer a unique model for adult chondrogenesis and cartilage repair and may serve as inspiration for novel cell-based therapies for skeletal pathologies, such as osteoarthritis.

## Introduction

Hyaline cartilage is a skeletal tissue that consists of a single cell type (the chondrocyte) embedded within a homogeneous, collagenous extracellular matrix (reviewed in *Gillis, 2018*). In mammals, hyaline cartilage is predominantly an embryonic tissue, making up the anlage of the axial (chondrocranial, vertebral and rib) and appendicular (limb) endoskeleton. The vast majority of mammalian hyaline cartilage is replaced by bone during the process of endochondral ossification, with cartilage persisting temporarily in epiphyseal growth plates, and permanently at relatively few sites within the adult skeleton (e.g. in joints, as articular cartilage – *Decker, 2017*). In juvenile mammals, growth of articular cartilage occurs by cellular rearrangement and increases in chondrocyte volume (*Decker et al., 2017*), and by appositional recruitment of new chondrocytes from a superficial population of slow-cycling chondroprogenitor cells (*Hayes et al., 2001*; *Dowthwaite et al., 2004*; *Karlsson et al., 2009*; *Williams et al., 2010*; *Kozhemyakina et al., 2015*). However, evidence for the presence of chondroprogenitor cells in adult articular cartilage is scant, and this – combined with the avascular nature of the tissue – may account for why mammalian articular cartilage cannot heal spontaneously following injury (*Hunziker, 1999*).

Chondrichthyans (cartilaginous fishes – sharks, skates, rays and holocephalans), on the other hand, possess an endoskeleton that is composed largely of hyaline cartilage, and that remains cartilaginous throughout life. Though chondrichthyans reinforce their endoskeleton with a superficial layer of calcified cartilage (in the form of small mineralized plates called 'tesserae' – *Dean and Summers, 2006*), the core of their endoskeletal elements persists as hyaline cartilage and does not undergo endochondral ossification. Like many fishes (and unlike mammals), chondrichthyans also exhibit an indeterminate type of growth, with a continued (albeit slow) increase in size through adulthood (*Dutta, 1994*; *McDowall, 1994*; *Frisk and Miller, 2006*). It therefore stands to reason that, in

*For correspondence: jag93@cam.ac.uk

Competing interests: The authors declare that no competing interests exist.

**eLife digest** For our joints to move around freely, they are lubricated with cartilage. In growing mammals, this tissue is continuously made by the body. But, by adulthood, this cartilage will have been almost entirely replaced by bone. It is also difficult for adult bodies to replenish what cartilage does remain – such as that in the joints.

When growing new cartilage, the body uses so-called progenitor cells, which have the ability to turn into different cell types. Progenitor cells are recruited to the joints, where they transform into cartilage cells called chondrocytes, which generate new cartilage. But adults lack these progenitor cells, leaving them unfit to heal damaged cartilage after injury or diseases like osteoarthritis.

In contrast, certain groups of fishes, such as skates, sharks and rays, produce cartilage throughout their life — indeed their whole skeleton is made of cartilage. So, what is the difference between these cartilaginous fishes and mammals? Why can they generate cartilage throughout their lives, while humans are unable to? And does this mean that these adult fish are better at healing injured cartilage?

Marconi et al. used skates (*Leucoraja erinacea*) to study how cartilage develops, grows and heals in a cartilaginous fish. Progenitor cells were found in a layer that wraps around the cartilage skeleton (called the perichondrium). These cells were also shown to activate genes that control cartilage development. By labelling these progenitor cells, their presence and movements could be tracked around the fish. Marconi et al. found progenitor cells in adult skates that were able to generate chondrocytes. Skates were also shown to spontaneously repair damaged cartilage in experiments where cartilage was injured.

Marconi et al. have identified the skate as a new animal model for studying cartilage growth and repair. Studying the mechanisms that skate progenitor cells use for generating cartilage could lead to improvements in current therapies used for repairing cartilage in the joints.

chondrichthyans, skeletal tissues may possess a persistent pool of chondroprogenitor cells to facilitate continued growth of their cartilaginous endoskeleton throughout adulthood, and that such cells (if present) could also impart the endoskeleton with an ability to undergo spontaneous repair following injury. However, basic mechanisms of hyaline cartilage development, growth and repair in chondrichthyans remain largely unexplored.

Here, we characterize the development and growth of the cartilaginous endoskeleton of a chondrichthyan, the little skate (*Leucoraja erinacea*), from embryonic development to adulthood. We demonstrate conservation of fundamental cellular and molecular characteristics of cartilage development between chondrichthyans and mammals, and we identify unique features of adult skate cartilage that contribute to its continued growth through adulthood. We further show that skates can repair surgically induced partial-thickness cartilage injuries, highlighting this system as a unique animal model for adult chondrogenesis and spontaneous hyaline cartilage repair.

## Results

### The metapterygium of the skate, *Leucoraja erinacea*

The pectoral fin endoskeleton of jawed vertebrates consisted ancestrally of three basal cartilages – from anterior to posterior, the propterygium, mesopterygium and metapterygium – and a series of articulating distal radials (*Davis et al., 2004*). Among extant jawed vertebrates, this ancestral 'tribasal' condition has been retained in the pectoral fins of chondrichthyans and non-teleost ray-finned fishes (e.g. in sturgeon, gar and bichir), but has been reduced in tetrapods and teleosts, to include only derivatives of the metapterygial and pro-/mesopterygial components, respectively (*Davis, 2013*). Our study focused on the metapterygium of the skate (*Figure 1*), as this element is relatively large, reliably identifiable across all embryonic and post-embryonic stages and easily accessible for surgical manipulation.

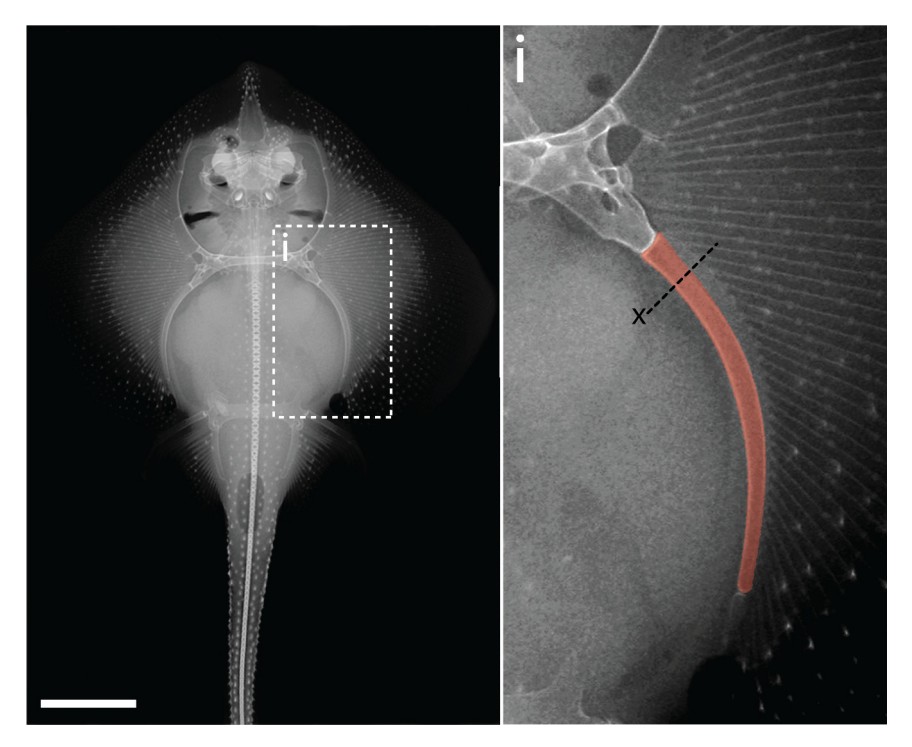

**Figure 1.** The metapterygium of the little skate, *Leucoraja erinacea*. A radiograph reveals the skeletal anatomy of an adult skate. The metapterygium (false colored red in **i**) is the caudalmost basal fin cartilage. The plane of section through the metapterygium used for *Figures 2–6* is indicated with a dashed line and x. Scale bar = 5 cm.

## Embryonic development and growth of cartilage in the skate metapterygium

In mammals, early cartilage development is marked by the accumulation of preskeletal mesenchyme into a 'condensation' at the site of future chondrogenesis (*Hall and Miyake, 2000*). Cells within this condensation begin to secrete cartilage extracellular matrix (ECM) components and undergo overt differentiation into chondrocytes. To investigate the early development of cartilage in the skate, we prepared a histological series of skate metapterygia from embryonic stage (S) 30 through to hatching and used a modified Masson's trichrome stain to visualize condensation, differentiation and ECM secretion.

At S30, the presumptive metapterygium exists as condensed mesenchyme, with pericellular Light Green staining (which appears blue) indicating onset of ECM secretion by cells within the condensation (*Figure 2a*). From S31-33 (*Figure 2b–d*), the metapterygium differentiates into cartilage and grows, with cells in the centre of the element adopting differentiated chondrocyte morphology (i.e. cells residing within ECM cavities or 'lacunae'), but cells in the periphery maintaining a less differentiated appearance. By hatching (*Figure 2e*), the cartilage of the metapterygium is surrounded by a distinct fibrous perichondrium (*Figure 2e*[i]), beneath which we begin to observe modifications of the ECM at sites of superficial calcification (i.e. developing tesserae – *Figure 2e*[ii]). By hatching, cells throughout the cartilage have adopted differentiated chondrocyte morphology, and are embedded within extensive collagenous ECM (*Figure 2f*).

## Conserved co-expression of genes encoding ECM components and upstream transcriptional regulators in embryonic skate chondrocytes

Mammalian hyaline cartilage ECM is composed largely of fibrils of type II collagen, which entrap aggregates of the hydrated proteoglycan aggrecan (*Eyre, 2002*; *Kiani et al., 2002*). *Col2a1* and *Agc* (the genes encoding type II collagen and aggrecan, respectively), in turn, are directly

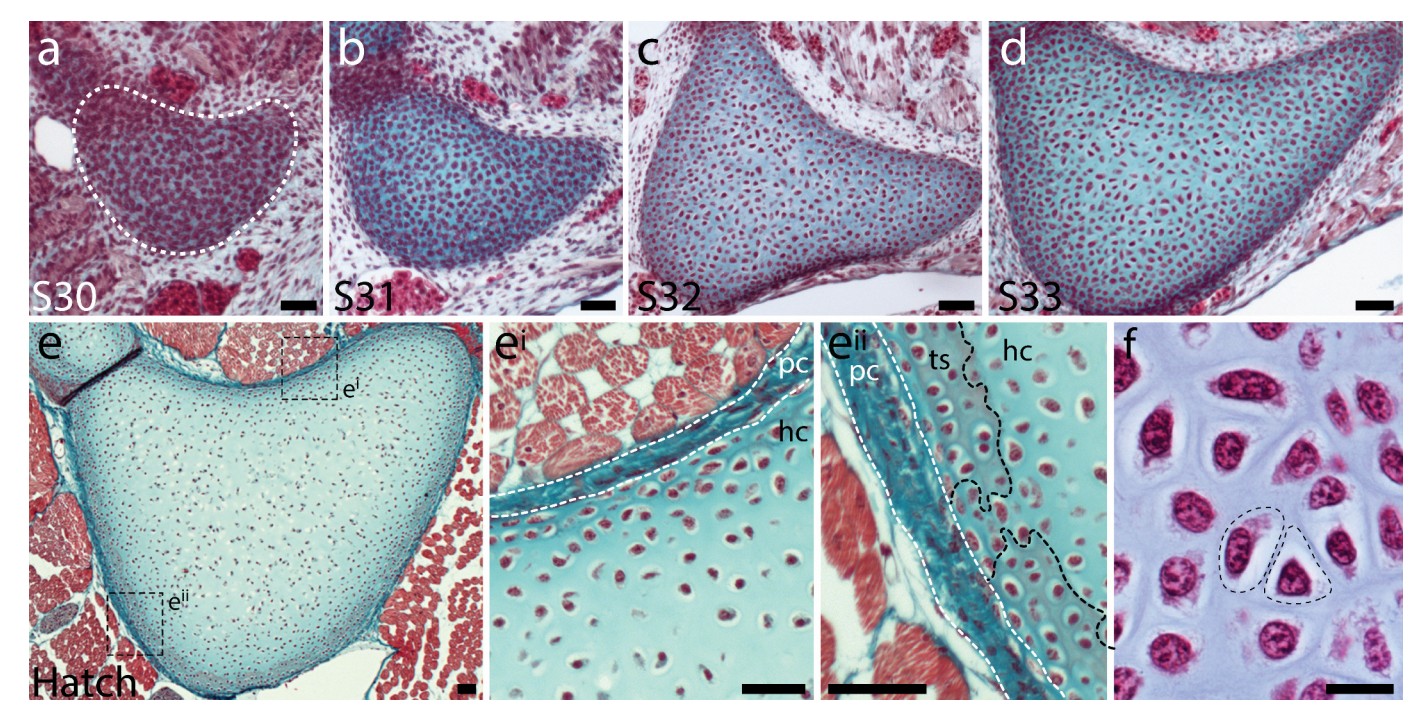

**Figure 2.** Embryonic development of the skate metapterygium. Transverse histological sections through the centre of the developing metapterygium at (**a**) S30 (white dashed line indicates condensation boundary), (**b**) S31, (**c**) S32, (**d**) S33 and (**e**) hatching. By hatching, the metapterygium is (**e**[i]) bound by a fibrous perichondrium, and (**e**[ii]) a surface layer of calcified cartilage in the form of mineralized 'tesserae' begins to develop beneath the perichondrium. (**f**) Cells in the hyaline cartilaginous core of the metapterygium adopt typical chondrocyte morphology, and are recessed within lacunae in the abundant extracellular matrix. All sections stained with modified Masson's trichrome. Plane of section as indicated in *Figure 1i*. *hc*, hyaline cartilage; *pc*, perichondrium; *ts*, tesserae. Scale bars: (**a-e**[ii]) 50 µm, (**f**) 20 µm.

transcriptionally regulated in chondrocytes by the SRY-box transcription factors Sox9, Sox5, and Sox6 (*Bell et al., 1997*; *Lefebvre et al., 1998*; *Lefebvre et al., 2001*). To test for conservation of these gene expression features in chondrocytes of the skate metapterygium, we characterized the co-expression of genes encoding cartilage ECM components and upstream transcriptional regulators in situ.

We first cloned fragments of skate *Cola2a1* (*Figure 3—figure supplement 1*) and *Agc* (*Figure 3—figure supplement 2*) and tested for their expression in the S32 metapterygium by chromogenic mRNA in situ hybridization. We found that both *Col2a1* (*Figure 3a*) and *Agc* (*Figure 3b*) are expressed in chondrocytes throughout the skate metapterygium, reflecting shared ECM properties between skate and mammalian hyaline cartilage. To test for conservation of the regulatory relationship between Sox5, Sox6 and Sox9, we used multiplexed fluorescent in situ hybridization by chain reaction (HCR) to test for co-expression of these genes (*Figure 3—figure supplements 3–4*) in metapterygium chondrocytes. We observed co-expression of *Col2a1* and *Sox9* in chondrocytes throughout the metapterygium (*Figure 3c–d*), as well as co-expression of *Col2a1*, *Sox5* and *Sox6* (*Figure 3e–f*), indicating likely conservation of regulation of genes encoding cartilage ECM components by SoxE- and SoxD-class transcription factors in skate cartilage.

## Proliferation of chondrocytes and putative perichondral progenitor cells in the metapterygium of skate hatchlings

To characterize patterns of cell proliferation within the growing metapterygium, we conducted a label retention experiment in skate hatchlings. Incorporation and detection of thymidine analogues, such as 5-ethynyl-2'-deoxyuridine (EdU), provides a sensitive readout of DNA synthesis and, by extension, cell proliferation (*Salic and Mitchison, 2008*). Briefly, hatchling skates were given a single intraperitoneal microinjection of EdU, and were then harvested at 1-, 5-, 10- and 40 days post-

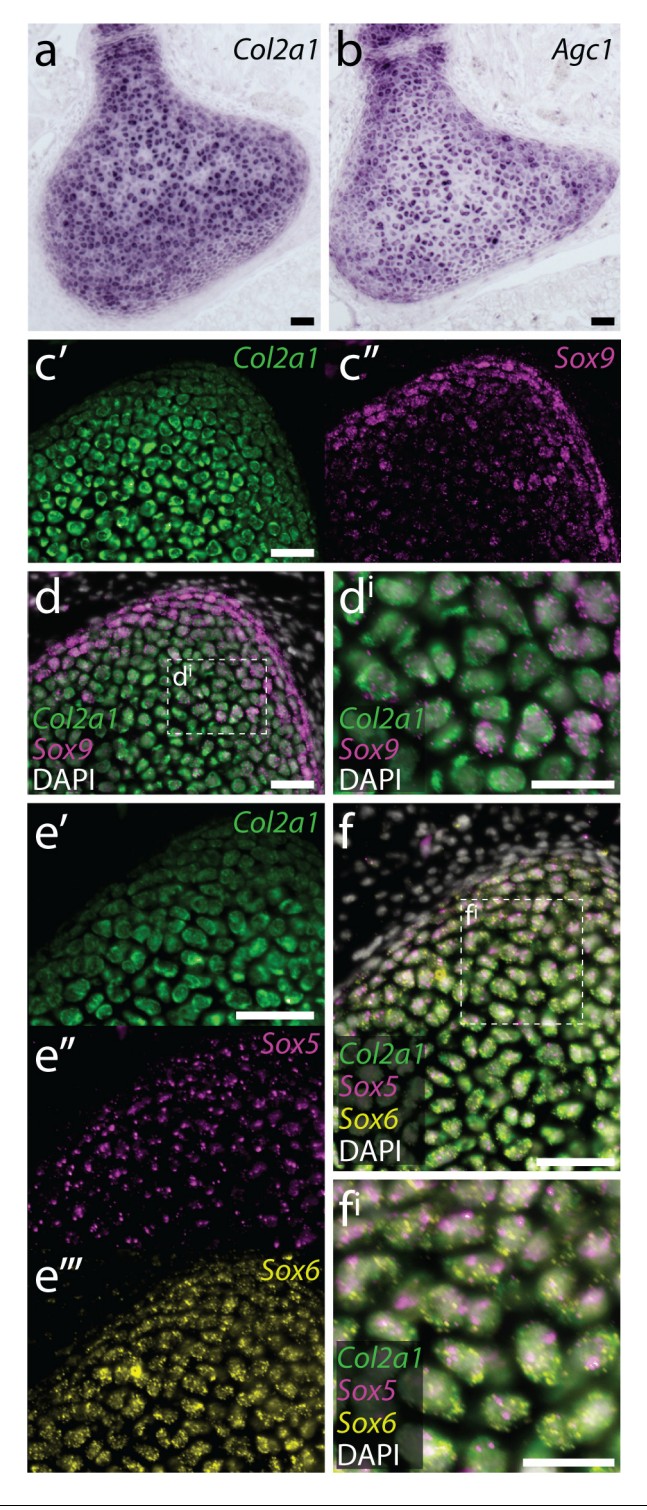

**Figure 3.** Conserved co-expression of genes encoding ECM components and upstream transcription factors in skate cartilage. (a) At S32, chromogenic mRNA in situ hybridization reveals that chondrocytes within the developing metapterygium express *Col2a1* and (b) *Agc1*. Multiplexed fluorescent mRNA in situ hybridization by chain reaction (HCR) reveals that skate chondrocytes co-expression (c-d) *Col2a1* and *Sox9*, and (e-f) *Col2a1*, *Sox5* and *Sox6*, pointing to conservation of transcriptional regulation of *Col2a1* by SoxD- and E-class transcription factors in jawed vertebrates. Plane of section as indicated in *Figure 1i*. Scale bars: (a-d) 50 μm, (dᶦ) 30 μm, (e-f) 50 μm, (fᶦ) 30 μm.

*Figure 3 continued on next page*

*Figure 3 continued*

The online version of this article includes the following figure supplement(s) for figure 3:

**Figure supplement 1.** Phylogenetic analysis of vertebrate fibrillar collagens.
**Figure supplement 2.** Phylogenetic analysis of vertebrate aggrecan.
**Figure supplement 3.** Phylogenetic analysis of the vertebrate SoxE family.
**Figure supplement 4.** Phylogenetic analysis of the vertebrate SoxD family.

injection (hereafter referred to as 1-, 5-, 10- and 40 day chase, respectively), to test for label retention within the metapterygium. In animals analyzed at 1 day chase, EdU+ chondrocytes were recovered throughout the cartilage of the metapterygium, though with a concentration of EdU+ cells around the periphery of the element (*Figure 4a*), pointing to the continued proliferation of differentiated chondrocytes at hatching. We also observed EdU+ cells within the perichondrium of the metapterygium at 1 day chase (*Figure 4b*). These label-retaining cells exhibited a distinct, flattened nuclear morphology (relative to adjacent chondrocytes), and were recovered in increasingly greater numbers within the perichondrium at 5- (*Figure 4c*) and 10 day chase (*Figure 4d*). By 40 days chase (*Figure 4d*), we observed a marked decrease in the number of EdU+ perichondral cells. This pattern of label retention (*Figure 4f*) is suggestive of an expanding or self-renewing cell population within the perichondrium, with a greater number of label-retaining cells arising through proliferation of EdU+ progenitors but reduction in the number of label-retaining cells with eventual dilution of EdU to undetectable levels. In animals analyzed at 10- (*Figure 4g*) and 40 days chase (*Figure 4h*), we observed numerous clusters of EdU+ chondrocytes in cartilage immediately adjacent to EdU+ perichondral cells. As cells of the inner perichondrium are known to give rise to new chondrocytes in the cartilaginous anlage of the chick limb skeleton (*Scott-Savage and Hall, 1979*), we speculate that these subperichondral EdU+ chondrocytes are the progeny of label-retaining perichondral cells, and that growth of the hatchling metapterygium occurs both through proliferation of differentiated chondrocytes, and by recruitment of new chondrocytes from progenitor cells within the perichondrium.

Using mRNA in situ hybridization by HCR, we investigated gene expression at the cartilage-perichondrium interface within the metapterygium of skate hatchlings. As in S32 embryos, we noted co-expression of *Col2a1* and *Sox9* (*Figure 4i*) and *Col2a1*, *Sox5* and *Sox6* (*Figure 4j*) in chondrocytes of the metapterygium. However, we also observed cells within the perichondrium that expressed *Sox9*, *Sox5* and *Sox6* but not *Col2a1* (*Figure 4i,j*). These cells were invariably located within the innermost region of the perichondrium, at the cartilage-perichondral interface, and exhibited the flattened nuclear morphology of the EdU+ perichondral cells described above. Taken together, our label retention and gene expression data point to the perichondrium as a source of cartilage progenitor cells in skate hatchlings.

## Histological features of cartilage in the adult skate metapterygium

We next examined histological features of the adult skate metapterygium, by vibrotomy (*Figure 5a*) and histochemical staining of paraffin sections with a modified Masson's trichrome stain (*Figure 5b*). In transverse sections through the metapterygium, we noted that the core of the element has the glassy appearance of hyaline cartilage, while the surface of the cartilage is covered by a rind of calcified tesserae (*Figure 5a–b*). Cells in the core of the metapterygium exhibit typical hyaline chondrocyte morphology (*Figure 5c*), are embedded in an ECM that stains with Light Green, and express high levels of type II collagen (*Figure 5d*). A higher magnification view of the tesserae reveals that these sit beneath a well-developed, fibrous perichondrium, which stains variably red and blue/green with trichrome staining (*Figure 5e*). We intermittently see a thin layer of unmineralized cartilage between the surface of the tesserae and the perichondrium, likely corresponding with the 'supratesseral' cartilage that has been previously reported in the stingray, *Urobatis halleri* (*Seidel et al., 2017*). Closer examination of the intertesseral joint region (i.e. the zone of cartilage between adjacent tesserae – *Figure 5f*) reveals cells with typical chondrocyte morphology within ECM up to the boundary between hyaline cartilage and the perichondrium, as well as a distinct population of flattened, spindle-shaped cells at the boundary between cartilage and perichondrium (*Figure 5f'*), hereafter referred to as 'inner perichondral cells'.

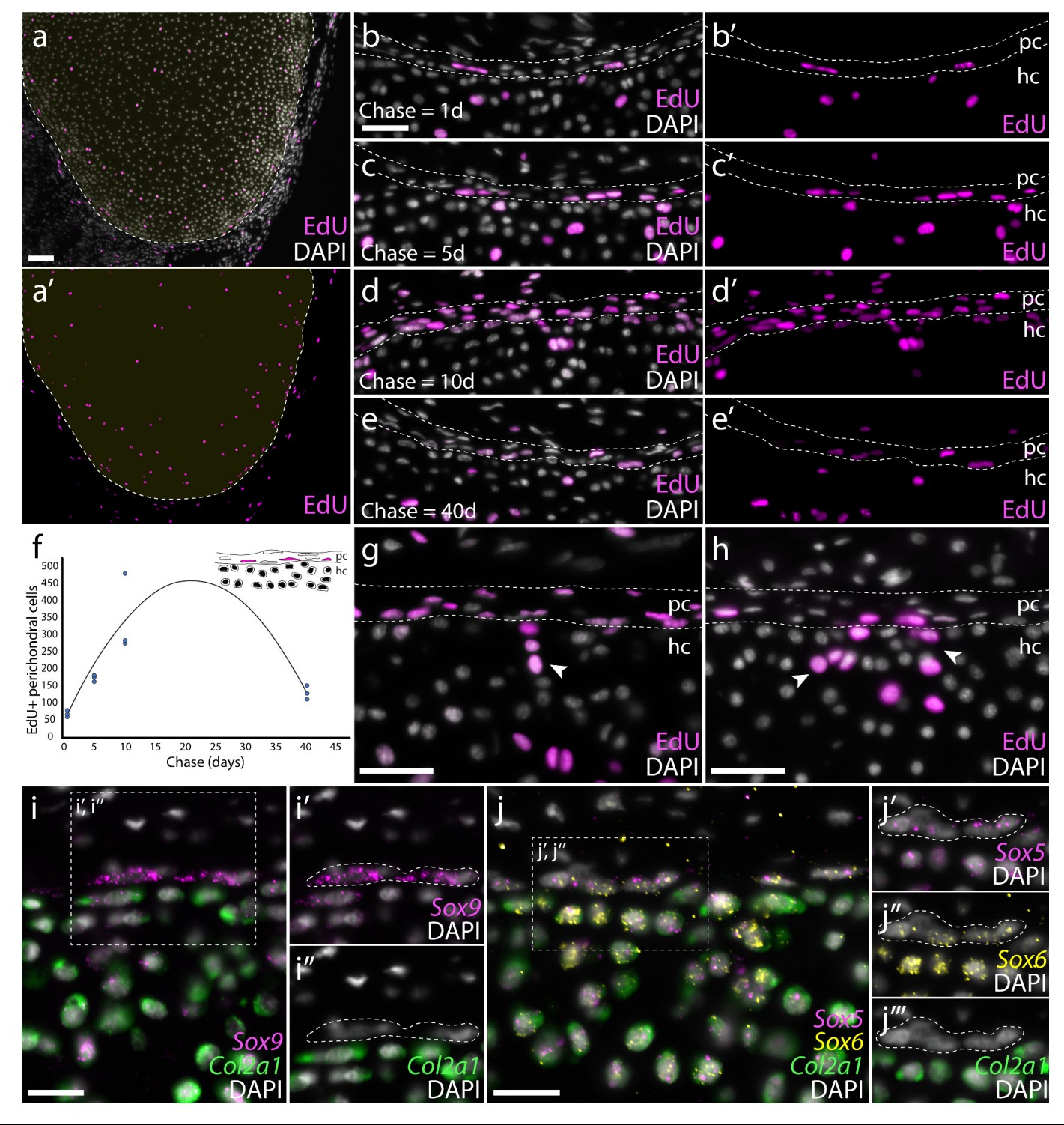

**Figure 4.** Proliferation of chondrocytes and recruitment of new chondrocytes from putative perichondral progenitors in skate hatchlings. (**a**) Transverse sections through the metapterygium of skate hatchlings reveals a concentration of label-retaining (EdU+) cells around the periphery of the element, both in the cartilage and in the perichondrium, as well as sparse label-retaining cells in the center of the metapterygium. An EdU pulse-chase experiment reveals that the perichondrium is an expanding cell population. (**b**) 1 day chase reveals sporadic labelling of cells within the perichondrium, but an increase in the number of EdU+ perichondral cells after (**c**) 5- and (**d**) 10 day chases. (**e**) After a 40 day chase, reduction in the number EdU+ cells in the perichondrium points to label dilution over several rounds of cell division. (**f**) Sum total EdU+ perichondral cells from three adjacent 8 μm transverse paraffin sections through the metapterygium after EdU pulse and 1-, 5-, 10- and 40 day chase (n = 3 individuals per time point). (**g**) The

*Figure 4 continued on next page*

*Figure 4 continued*

occurrence of clusters of EdU+ chondrocytes immediately adjacent to EdU+ perichondral cells after 10 day and (h) 40 day chase points to the likely perichondral origin of these cells. mRNA in situ hybridisation by HCR for (i) *Sox9* and *Col2a1*, and for (j), *Sox5, Sox6* and *Col2a1* reveals a population of perichondral cells that sit at the cartilage-perichondral boundary, and that co-express *Sox9, Sox5* and *Sox6* but not *Col2a1*. These cells (white dashed outline) are morphologically similar to the label-retaining perichondral cells identified in (b-e). In all images, plane of section as indicated in *Figure 1i*. *hc*, hyaline cartilage; *pc*, perichondrium. Scale bars: (a) 100 μm, (b-e), (g-h) 50 μm, (i-j) 25 μm.

Interestingly, the adult skate metapterygium is permeated by a series of canals, which originate at the surface of the cartilage, and extend toward the core of the element (*Figure 5a–b*). These cartilage canals occur throughout the metapterygium, originate within the perichondrium and enter the cartilage between tesserae (i.e. through the intertesseral joint region), and are not lined by an epithelium (*Figure 5g*). Cartilage canals contain an abundance of cells, including some red blood cells (*Figure 5h*), but predominantly cells with connective tissue/mesenchymal morphology – many of which appear to be invading from the canal into surrounding cartilage ECM (*Figure 5i*). Immunostaining for type II collagen reveals that cartilage canals are zones of active ECM synthesis, with high levels of type II collagen being secreted by cells at the periphery of the canals (*Figure 5j*).

## Conserved co-expression of *Col2a1*, *Sox5*, *Sox6* and *Sox9* in peripheral chondrocytes and perichondral cells in adult skate cartilage

To test for cells that are actively expressing cartilage ECM products in the adult skeleton, we analyzed expression of *Col2a1* by mRNA in situ hybridization on sections of adult metapterygium. High levels of *Col2a1* transcription were detected around the periphery of the cartilage – that is in the intertesseral joint region and in cartilage adjacent to tesserae – and also in the thin layer of supratesseral cartilage that sits between the tesserae and the perichondrium (*Figure 6a*). In situ hybridization by HCR revealed that both supratesseral and peripheral chondrocytes co-expressed *Col2a1* and *Sox9* (*Figure 6b–c*), as well as *Col2a1, Sox5, Sox6* (*Figure 6d–e*), pointing to retention of transcriptional features of developing cartilage around the periphery of the adult metapterygium. Interestingly, as in hatchling skates, we also observed co-expression of *Sox5, Sox6* and *Sox9* (but not *Col2a1*) in the flattened cells of the inner perichondrium (*Figure 6b,d*).

## Label-retaining progenitor cells in adult perichondrium give rise to new chondrocytes during cartilage growth

Given the indeterminate growth of cartilaginous fishes, we speculated that the transcriptional signature of embryonic cartilage in the periphery of the adult skate metapterygium could reflect recently born chondrocytes contributing to ECM expansion, while the presence of this signature in the inner perichondrium could reflect progenitors of new chondrocytes. We therefore sought to test for the presence and fate of cycling cells in the adult skate metapterygium using a label retention experiment. Due to the relatively slow growth rate of cartilaginous fishes, we reasoned that a pulse-chase label retention experiment could be used not only to localize cell proliferation within the metapterygium, but also to lineage trace label-retaining cells to test for contributions to hyaline cartilage. Briefly, eight adult female skates were given three intraperitoneal injections of EdU, 48 hr apart, and two animals were then euthanized, fixed and processed for EdU detection 3 days, 1 month, 2 months and 5.5 months following the final IP EdU injection (hereafter referred to as 3 day, 1 month, 2 month and 5.5 month chase, respectively). EdU detection was performed on transverse paraffin sections though the metapterygium, with EdU+ cells scored according to their location in the outer perichondrium, inner perichondrium, cartilage canals or cartilage (i.e. chondrocytes). 30–40 sections were analyzed from each animal, but quantification was performed by counting the sum total and tissue localization of EdU+ cells in five adjacent sections through the metapterygium (as indicated in *Figure 1i*) after pulse + 3 days, 1 month, 2 month and 5.5 month chase (*Table 1*).

After a 3 day chase, EdU+ cells were recovered almost exclusively in the perichondrium, with most appearing as cells with rounded nuclei in the outer perichondrium (*Figure 7a*), and relatively few as flattened cells of the inner perichondrium (*Figure 7b*). No EdU+ chondrocytes were detected after a 3 day chase. After 1 month and 2 month chases, we continued to detect EdU+ cells within the outer and inner perichondrium, and we also observe EdU+ cells within the cartilage canals that originate in the perichondrium and permeate the core of the metapterygium (*Figure 7c–d*). After a

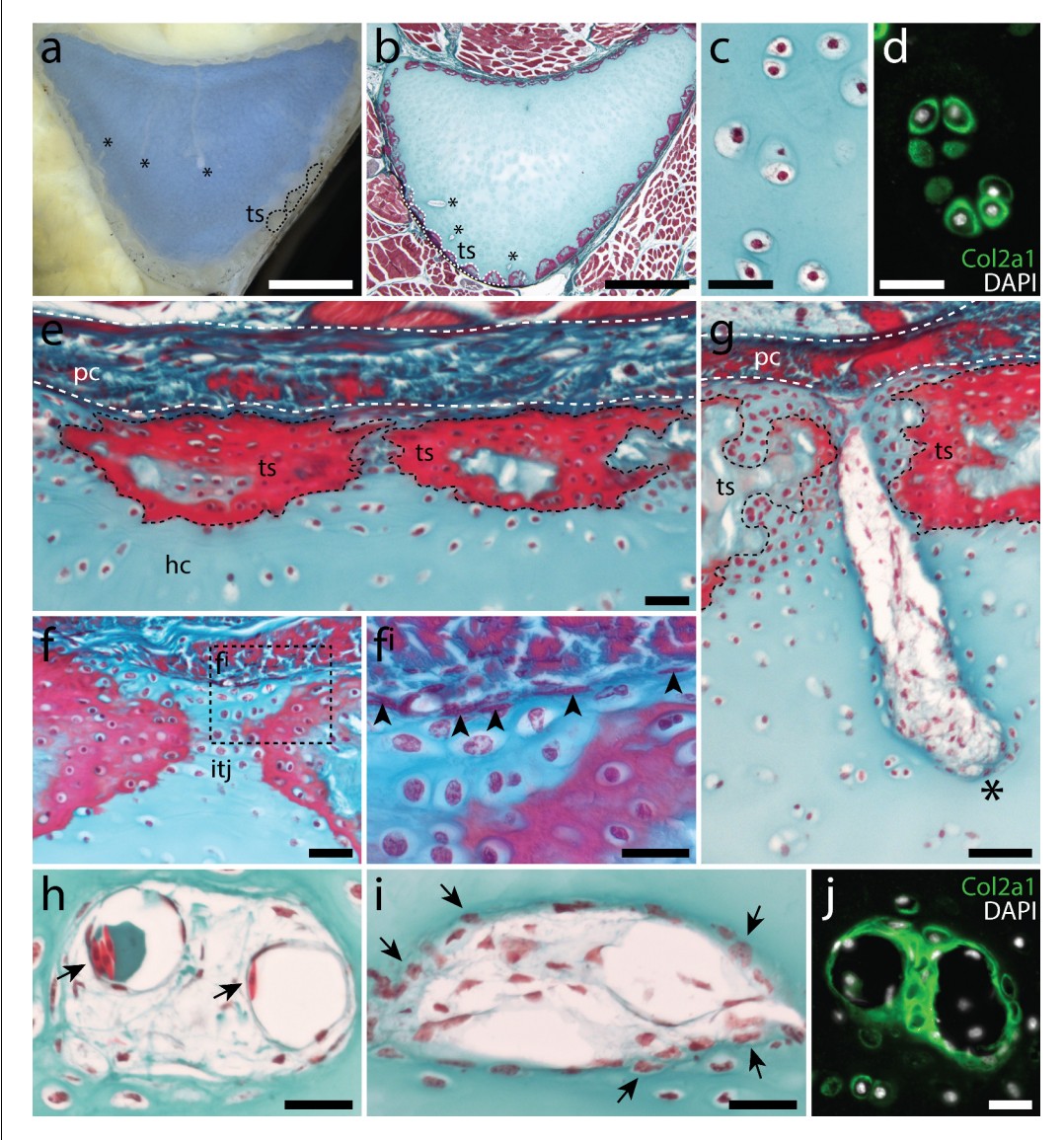

**Figure 5.** Histological features of the metapterygium in the adult skate. (a) Transverse vibratome and (b) histological sections through the adult skate metapterygium reveal cartilage canals (asterisks) originating in the perichondrium and extending into the cartilaginous core of the element. The surface of the metapterygium is covered by calcified tesserae (dashed outlines). (c) Cells within the hyaline cartilage core of the metapterygium exhibit typical chondrocyte morphology, and (d) are surrounded by abundant pericellular type II collagen. (e) Mineralized tesserae sit between the hyaline cartilage core and an overlying fibrous perichondrium. (f) Examination of the unmineralized hyaline cartilage of the intertesseral joint region reveals a population of flattened, spindle-shaped cells (black arrowheads in f^i) sitting at the boundary between the cartilage and the perichondrium. (g) Cartilage canals (asterisk) can be seen entering the hyaline cartilage of the metapterygium through the intertesseral joint region. These canals originate in the perichondrium, and extend into the core cartilage of the metapterygium. (h) Cartilage canals are not lined by an epithelium, and contain some red blood cells (black arrows), but predominantly (i) connective tissue-like cells – many of which appear to be invading adjacent cartilage ECM (black arrows). (j) Cartilage canals are sites of active type collagen secretion, as indicated by positive immunostaining for Col2a1. (b-c) and (e-i) stained with modified Masson's trichrome. Plane of section as indicated in *Figure 1i*. *hc*, hyaline cartilage; *itj*, intertesseral joint region; *pc*, perichondrium; *ts*, tesserae. Scale bars: (a-b) 2 mm, (c-d) 30 μm, (e-f) 50 μm, (f^i) 30 μm, (g) 50 μm, (h-j) 30 μm.

5.5 month chase, we detected an abundance of EdU+ cells in both the outer and inner perichondrium (*Figure 7e*), as well as abundant EdU+ chondrocytes in the peripheral hyaline cartilage of the intertesseral joint region (*Figure 7f*) and in cartilage subjacent to the tesserae (*Figure 7g*), as well as relatively few EdU+ chondrocytes within the mineralized matrix of the tesserae (*Table 1*). EdU+ cells are present in greater abundance within cartilage canals after a 5.5 month chase (*Figure 7h*) and

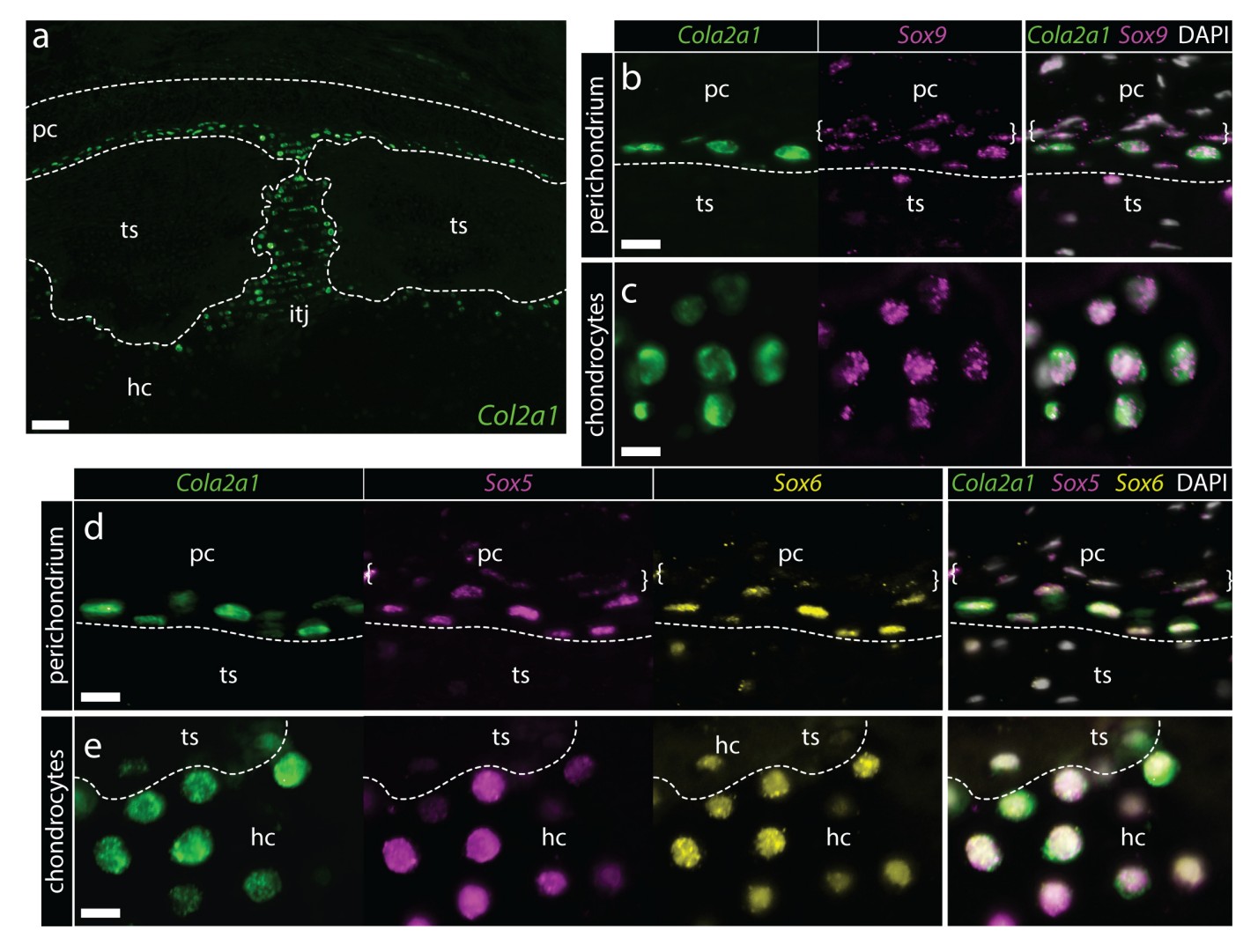

**Figure 6.** Conserved co-expression of *Col2a1*, *Sox9*, *Sox5* and *Sox6* in peripheral chondrocytes and inner perichondral cells of the adult metapterygium. (**a**) *Col2a1* is highly expressed by peripheral chondrocytes of the metapterygium, in supratesseral chondrocytes and in the hyaline cartilage around the tesserae (**b**) Supratesseral and (**c**) peripheral chondrocytes co-express *Col2a1* and *Sox9*, as well as (**d-e**) *Col2a1*, *Sox5* and *Sox6*. In (**b**) and (**d**) white brackets indicate inner perichondral cells that co-express *Sox9*, *Sox5* and *Sox6* but not *Col2a1*. Plane of section as indicated in *Figure 1i*. *hc*, hyaline cartilage; *pc*, perichondrium; *ts*, tesserae. Scale bars: (**a**) 100 µm, (**b-e**) 15 µm.

given the relative dearth of EdU+ cells in cartilage canals after the 3 day, 1 month and 2 month chases, we speculate that these EdU+ cartilage canal cells are of perichondral origin. After the 5.5 month chase, we also observed EdU+ cells that appeared to be invading hyaline cartilage from the blind ends of cartilage canals (*Figure 7i*), and in one individual we observed three instances of EdU+ chondrocytes immediately adjacent to the blind ends of cartilage canals (*Figure 7j*).

Taken together, our label retention and gene expression data point to the morphologically distinct *Sox5/Sox6/Sox9* + cells of the inner perichondrium as adult cartilage progenitor cells, with a capacity to give rise to new chondrocytes both in the periphery (i.e. appositional growth), and also in the core of the metapterygium, via cartilage canals (i.e. interstitial growth). Additionally, while the number of EdU+ cells is highly variable between individuals, the general trend of a greater number of label-retaining perichondral cells after the 2 month and 5.5 month chases – and, more specifically, a greater number of label-retaining cells within the inner perichondrium – points to the likely self-renewal of perichondral cells, perhaps with a progressive sequence of differentiation from outer

**Table 1.** Recovery of EdU-retaining cells within the metapterygium of adult skates after pulse and 3 day, 1-, 2- and 5.5 month chase.

| Chase time | Number of EdU-retaining cells in... | | | |
| --- | --- | --- | --- | --- |
| | Perichondrium (outer) | Perichondrium (inner) | Cartilage canals | Chondrocytes |
| 3 days (1) | 22 | 3 | 1 | 0 |
| 3 days (2) | 6 | 0 | 0 | 0 |
| 1 month (1) | 9 | 7 | 3 | 0 |
| 1 month (2) | 9 | 0 | 2 | 0 |
| 2 months (1) | 16 | 0 | 2 | 2 |
| 2 months (2) | 107 | 31 | 4 | 0 |
| 5.5 months (1) | 271 | 259 | 20 | 25 |
| 5.5 months (2) | 66 | 15 | 20 | 10 |

perichondral cell to inner perichondral cell, and eventually to chondrocyte – either in the periphery, or deeper in the core (via cartilage canals).

## Adult skates spontaneously repair partial-thickness cartilage injury

Mammalian articular hyaline cartilage is unable to spontaneously heal following injury. Rather, articular cartilage injuries tend to infill with fibrocartilage – a subtype of cartilage that exhibits large bundles of collagen fibres, and with ECM composed substantially of type I collagen (*Eyre and Wu, 1983*; *Benjamin and Ralphs, 2004*). Fibrocartilage is mechanically inferior to hyaline cartilage at the articular surfaces of synovial joins, and its formation within articular cartilage lesions can result in the onset of degenerative osteoarthritis. We sought to test whether the persistence of adult chondrogenesis in the skate metapterygium – and the presence of cartilage progenitor cells in the perichondrium – correlated with an ability to spontaneously repair injured hyaline cartilage. We conducted a surgical cartilage injury experiment, in which a metapterygium cartilage biopsy was performed in 26 adult skates using a 4 mm biopsy punch (producing a cartilage void of ~1/4 to 1/3 diameter of the metapterygium – *Figure 8a*). Two animals were euthanized one-week post-operation, and at monthly intervals for the following year, and processed histologically to assess the extent of repair.

In animals assessed at 1 and 2 months post-operation (mpo) (n = 4), cartilage injuries were infilled with a fibrous connective tissue (*Figure 8b, b^i*), but by 3mpo (n = 2), this connective tissue began to differentiate into cartilage (i.e. with typical hyaline chondrocytes, albeit at a much higher density than in the adjacent, uninjured cartilage – *Figure 8c, c^i*). In animals assessed at 4-10mpo (n = 14), tissue within the injury sites showed varying degrees of progressive differentiation into cartilage, starting from the interface between the injury site and adjacent cartilage and progressing toward the surface of the metapterygium (*Figure 8—figure supplement 1*), and by 11-12mpo (n = 4), injury sites were completely or near-completely filled with repair cartilage (*Figure 8d*, *Figure 8—figure supplement 2*, *3*). Chondrocytes within the repair cartilage remained at much higher density than in the adjacent cartilage, and the surface of the injury site remained irregular, with some superficial red staining of the ECM (*Figure 8d^i*). This could reflect the re-appearance of tissue with a perichondral-like nature, or a step toward re-establishment of tesserae at the surface of the metapterygium, as these tissues stained variably red-blue and red, respectively, with modified Masson's trichrome stain. However, the vast majority of repair tissue resembled typical hyaline cartilage, with a Light Green-stained ECM that integrates with adjacent uninjured cartilage and no evidence of ECM fibre bundles typical of fibrocartilage (*Figure 8d^ii*).

We tested whether the ECM of repair cartilage was composed of type II collagen (as in typical hyaline cartilage) or a mixture of types I and II collagen (as in fibrocartilage) using immunofluorescence. We observed strong, positive staining for type II collagen throughout the ECM of repair cartilage, as well as pericellular staining for type II collagen in adjacent uninjured hyaline cartilage (*Figure 8e* – *Figure 8—figure supplement 4*). Conversely, we observed no positive staining for type I collagen in repair or uninjured cartilage (including cartilage canals), despite strong positive staining in adjacent skeletal muscle fibres (*Figure 8—figure supplement 4*). These findings suggest that

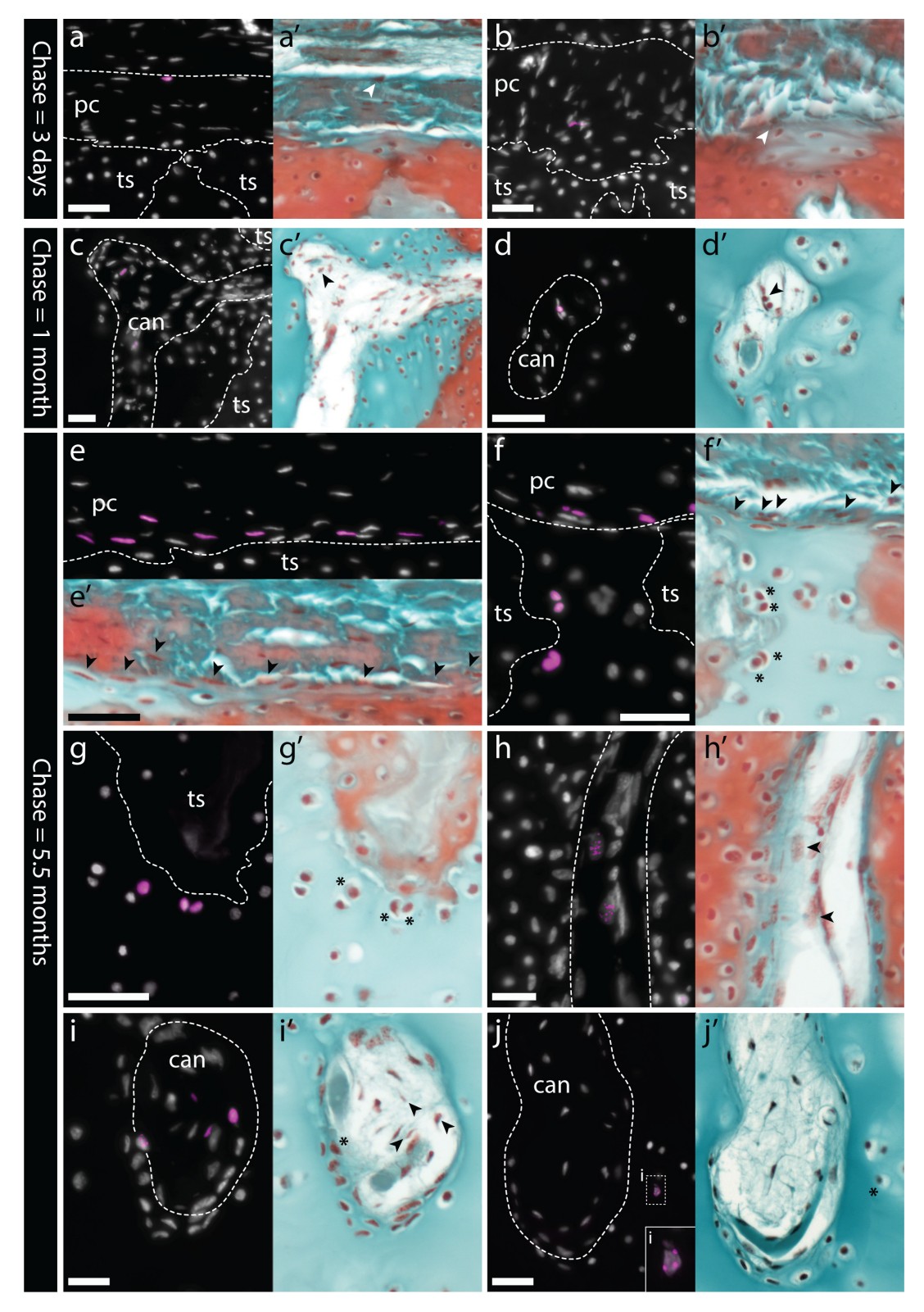

**Figure 7.** Label-retaining perichondral cells are cartilage progenitors in the adult metapterygium. (a) After a 3 day chase, EdU retention is detected in cells of the outer and (b) inner perichondrium. (c-d) After a 1 month chase, EdU-retaining cells are additionally detected inside cartilage canals. (e) After a 5.5 month chase, EdU-retaining cells are detected in abundance in the inner perichondrium, and also in peripheral chondrocytes, including (f) in the hyaline cartilage of the intertesseral joint region and (g) in hyaline cartilage beneath tesserae. (h) EdU-retaining cells are detected in greater abundance

*Figure 7 continued on next page*

*Figure 7 continued*

inside cartilage canals, and (i) can also be seen migrating from inside cartilage canals into adjacent ECM. (j) EdU-retaining chondrocytes are also detected in the core of the metapterygium, adjacent to the blind end of cartilage canals. For each panel, the same section was imaged for EdU detection (counterstained with DAPI), and subsequently stained with modified Masson's trichrome. In histochemical images, EdU+ nuclei in the perichondrium or in cartilage canals are indicated with arrowheads, while EdU+ chondrocytes are indicated with an asterisk. Plane of section as indicated in *Figure 1i. can*, canal; *pc*, perichondrium; *ts*, tesserae. Scale bars: (**a-c**) 50 μm, (**d**) 30 μm, (**e-g**) 50 μm, (**h-j**) 30 μm.

adult skate repair cartilage produces ECM similar to that of adjacent hyaline cartilage and is unlike the fibrocartilaginous repair tissue that typically fills mammalian chondral defects.

Interestingly, in two animals collected between 4-10mpo, our biopsy had been unsuccessful, with the biopsy punch perforating the surface of the metapterygium but failing to remove a wedge of cartilage. In both cases, a large mass of ectopic cartilage formed above the tesserae on the surface of the metapterygium, but beneath the fibrous perichondrium (*Figure 8—figure supplement 5*). This suggests that mechanical perturbation of the perichondrium may act as an inductive cue for onset of a chondrogenic injury response, and is consistent with the perichondrium as a source of new cartilage not only during normal adult growth, but also following injury.

## Discussion

The cartilaginous nature of the endoskeleton in sharks, skates, rays and holocephalans has long been appreciated, but the fundamental processes of cartilage development and growth in this group remain largely unexplored. We have shown that embryonic development of cartilage in the skate closely mirrors development of mammalian cartilage, but that unique features of the adult skate skeleton – including the presence of chondroprogenitors in the perichondrium and a network of cartilage canals – facilitates the continued growth of cartilage throughout life. This persistence of adult chondrogenesis, in turn, correlates with ability to spontaneously repair injured cartilage, in a manner so far undocumented in any other vertebrate taxon.

### Evolution of endoskeletal development in jawed vertebrates

The endoskeleton of bony fishes (including tetrapods) forms largely through a process of endochondral ossification. In endochondral ossification, pre-skeletal mesenchyme aggregates to form a condensation at the site of skeletogenesis, and cells within this condensation undergo progressive differentiation, starting from the centre, into chondrocytes and eventually enlarged, hypertrophic chondrocytes (*Hall, 2005*; *Karsenty et al., 2009*; *Long and Ornitz, 2013*). Chondrocyte and hypertrophic chondrocyte fates are determined and characterized by the expression of genes encoding distinct sets of transcription factors and ECM components, with the transcription factors Sox5, Sox6 and Sox9 regulating the expression of *Col2a1* and *Agc1* in chondrocytes (*Bell et al., 1997*; *Lefebvre et al., 1998*; *Lefebvre et al., 2001*; *Smits et al., 2001*; *Akiyama et al., 2002*; *Han and Lefebvre, 2008*), and the transcription factor Runx2 regulating the expression of *Col10a1* (the gene encoding non-fibrillar type X collagen) in hypertrophic chondrocytes (*Linsenmayer et al., 1991*; *Takeda et al., 2001*; *Zheng et al., 2009*; *Simões et al., 2006*; *Higashikawa et al., 2009*; *Ding et al., 2012*). Hypertrophic cartilage is ultimately invaded by vasculature, and is replaced by bone, with bone-forming cells (osteoblasts) arising both from the perichondrium/periosteum, and through transdifferentiation of hypertrophic chondrocytes (*Colnot et al., 2004*; *Roach, 1992*; *Roach et al., 1995*; *Zhou et al., 2014*; *Yang et al., 2014*; *Park et al., 2015*; *Hu et al., 2017*). Within growing endochondral bones, non-hypertrophic cartilage persists in the growth plate, where chondrocytes continue to proliferate and contribute to lengthening of an element, and at points of endoskeletal articulation. Upon cessation of growth, growth plate cartilage will hypertrophy and ossify, with non-hypertrophic cartilage persisting only at articular surfaces.

In skate, onset of endoskeletal development is marked by the appearance of mesenchymal condensations at sites of skeletogenesis, and cells within condensations differentiate into chondrocytes progressively, from the centre of the condensation to the periphery. This differentiation of condensed mesenchyme is accompanied by co-expression in chondrocytes of *Sox5*, *Sox6*, *Sox9*, *Col2a1* and *Agc1*, pointing to conservation of the regulatory interaction between SoxD- and E-class transcription factors and the genes encoding type II collagen and aggrecan in cartilaginous and bony

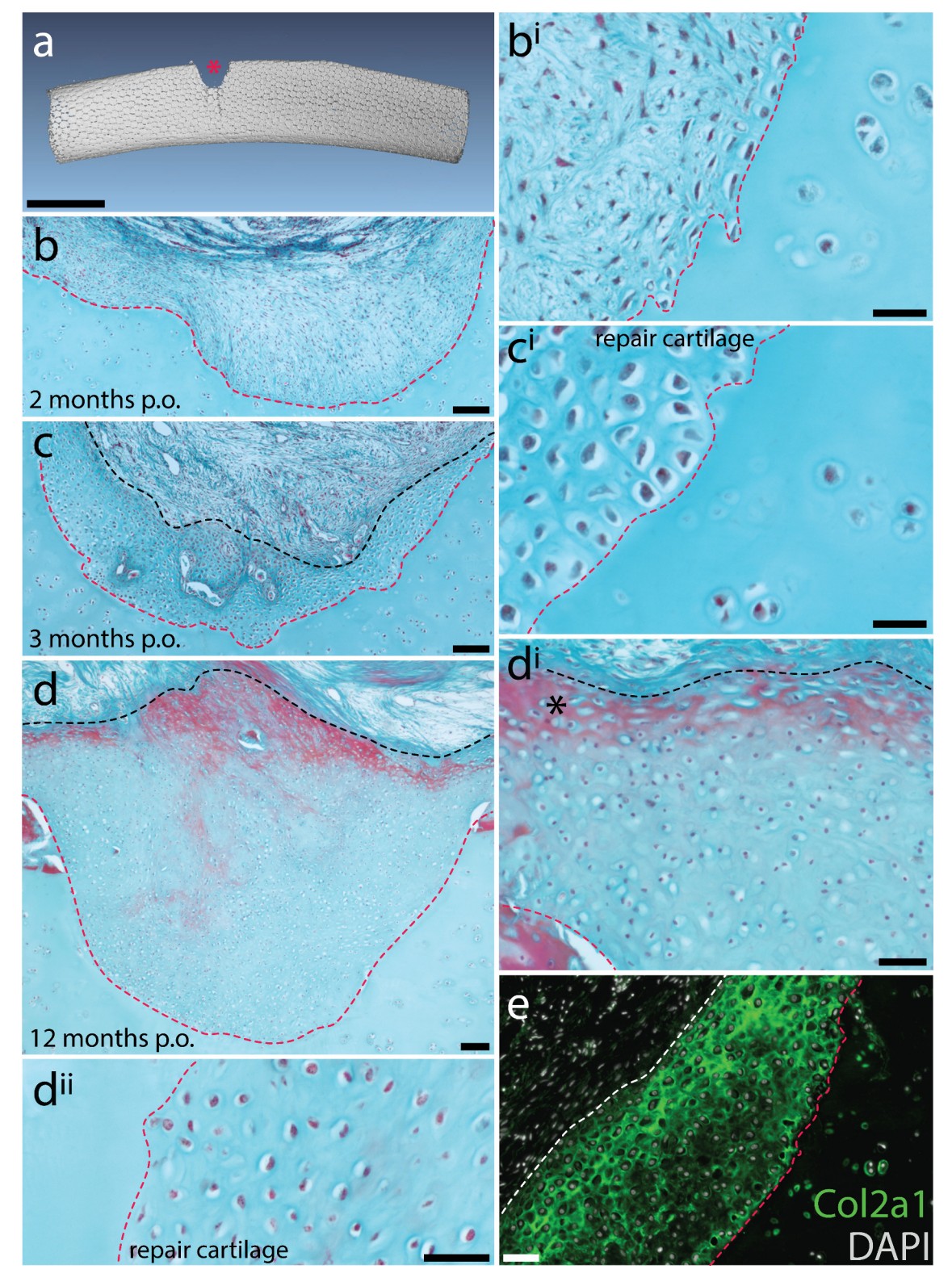

**Figure 8.** Spontaneous repair of hyaline cartilage in the skate. (**a**) 3D reconstruction of a dissected metapterygium 2 weeks following experimental cartilage injury. Note the biopsy (red asterisk) has left a void of ~1/3 the diameter of the metapterygium. (**b, b^i**) By 2 months post-operation (p.o.), the injury site has been filled with a fibrous connective tissue, and (**c, c^i**) by 3 months p.o., this connective tissue begins to differentiate into cartilage. Note that the cells of the repair tissue adopt chondrocyte morphology, and the ECM of the repair tissue is integrated with adjacent cartilage. (**d**) By 12

*Figure 8 continued on next page*

*Figure 8 continued*

months p.o., the injury site has been completely filled with repair cartilage. (d$^i$) Red staining of ECM at the surface of the repair tissue (*) could indicate the re-appearance of tissue with a perichondral-like nature, or the re-establishment of tesserae at the injured surface of the metapterygium. However, (d$^{ii}$) the vast majority of repair tissue is composed of typical hyaline cartilage. (e) Immunofluorescence reveals abundant type II collagen (Col2a1) in the ECM of repair cartilage. In (b-d), the red dashed line indicates the boundary of the biopsy, and the black dashed line indicates the extent of repair cartilage. In (e) the red dashed line indicates the boundary of the biopsy, and the white dashed line indicates the extent of repair cartilage. *hc*, hyaline cartilage; *pc*, perichondrium; *ts*, tesserae. Scale bars: (a) 1 cm, (b) 100 μm, (b$^i$) 30 μm, (c) 100 μm, (c$^i$) 30 μm, (d) 100 μm, (d$^i$) 50 μm, (d$^{ii}$) 50 μm, (e) 50 μm. The online version of this article includes the following figure supplement(s) for figure 8:

**Figure supplement 1.** Cartilage repair in the metapterygium of the skate 8 months post-operation.
**Figure supplement 2.** Cartilage repair in the metapterygium of the skate 11 months post-operation.
**Figure supplement 3.** Cartilage repair in the metapterygium of the skate 12 months post-operation.
**Figure supplement 4.** Collagen composition of ECM in uninjured and repair cartilage in the skate metapterygium 8 months post-operation.
**Figure supplement 5.** Ectopic cartilage following perichondral disruption in the metapterygium of the skate.

fishes, and to the broad developmental and biochemical comparability of cartilage between these major vertebrate lineages.

Unlike in bony fishes, however, chondrocytes in the skate metapterygium do not undergo hypertrophy, but rather remain terminally differentiated in a non-hypertrophic state. While cartilaginous fishes, strictly speaking, lack bone, they nevertheless possess the vast majority of transcription factors and ECM components required to make bone (*Venkatesh et al., 2014*), and there are instances of mineralization within the skeleton of cartilaginous fishes that share molecular properties with bone – for example expression of *Col10a1*, *Col1a1* and *SPARC* in the mineralized areolar tissue of the vertebral column, and immunolocalization of types I and X collagen to tesserae and pre-mineralized supratesseral cartilage (*Egerbacher et al., 2006*; *Enault et al., 2015*; *Criswell et al., 2017*; *Seidel et al., 2017*; *Debiais-Thibaud et al., 2019*). These observations, combined with palaeontological evidence for the presence of bone along the gnathostome stem (*Donoghue et al., 2006*; *Charest et al., 2018*), are consistent with bone as an ancestral feature of jawed vertebrates, the loss of this tissue in extant cartilaginous fishes, and the independent re-deployment of deeply conserved mechanisms of cartilage mineralization at sites such as tesserae and the axial column.

## Adult chondroprogenitor cells contribute to post-embryonic growth of cartilage in the skate

There has been relatively little work on postembryonic growth of hyaline cartilage in cartilaginous fishes, perhaps owing to the general difficulties of locating and maintaining a suitable range of sub-adult life stages. Growth of the tessellated calcified cartilage of cartilaginous fishes has been well documented, particularly in the stingray *Urobatis halleri*, where it has been shown that tesserae continue to grow throughout life, and that this growth likely occurs by accretion, with continuous mineralization of a thin layer of 'supratesseral' cartilage that sits between the tesserae and the perichondrium, and of 'subtesseral' cartilage beneath the tesserae (*Dean et al., 2009*; *Seidel et al., 2016*; *Seidel et al., 2017*). We have discovered a population of label-retaining cartilage progenitor cells (characterized by co-expression of *Sox5*, *Sox6* and *Sox9*) in the inner perichondrium of adult skates, and we have traced their chondrocyte progeny to the peripheral unmineralized cartilage and (occasionally) mineralized tesserae of the metapterygium. Based on our observation of a similar cell type in the metapterygium perichondrium of hatchling skates, these observations point to a general mechanism of appositional post-embryonic cartilage growth, wherein progenitor cells of perichondral origin give rise to new chondrocytes in the periphery of the metapterygium – some of which will become incorporated into growing tesserae, while others (e.g. in the intertesseral joint regions) will persist as unmineralized hyaline cartilage (*Figure 9*).

In addition to appositional growth of cartilage in the metapterygium, we also find some evidence of interstitial growth, by the addition of new chondrocytes to the unmineralized cartilaginous core of the metapterygium. In our pulse-chase label retention experiments, EdU+ cells are only recovered in the perichondrium after a pulse + 3 day chase (with the exception of a single label-retaining cell in a cartilage canal), with no label-retention in differentiated chondrocytes. However, we observe a striking increase in label-retaining cells within cartilage canals and core cartilage after pulse + 1–5.5 month chases, including a small number of chondrocytes in the centre of the metapterygium

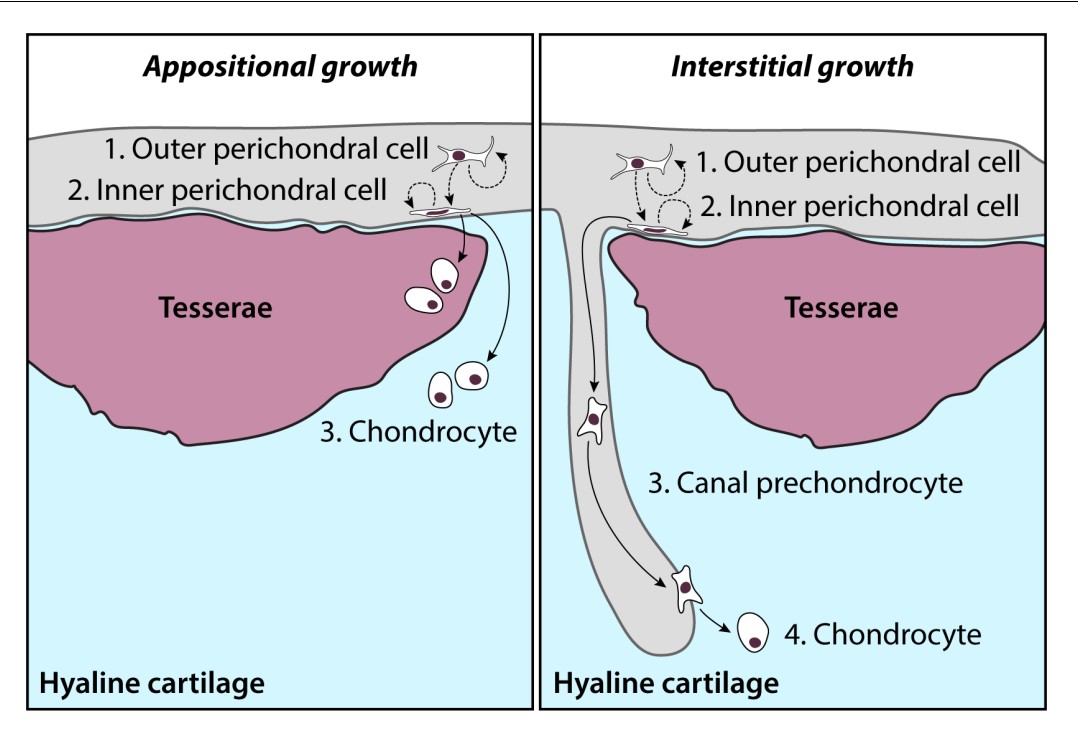

**Figure 9.** Model of adult chondrogenesis in the skate. Adult skate cartilage grows both appositionally (i.e. through peripheral addition of new chondrocytes and expansion of ECM) and interstitially (i.e. through incorporation of new chondrocytes deeper in the hyaline cartilage core). In both cases, progenitor cells reside within the perichondrium. Label-retention experiments indicate that cells cycle in both the outer and inner perichondrium (possibly with the former giving rise to the latter, along with self-renewal), and that cells of perichondral origin give rise to new chondrocytes. In appositional growth, chondrocyte progeny of perichondral progenitors resides in peripheral hyaline cartilage (or, occasionally, become incorporated into growing tesserae). In interstitial growth, prechondrocyte progeny of perichondral progenitors is transported to the cartilage core, where they ultimately invade the cartilage ECM and differentiate into chondrocytes.

cartilage core, immediately adjacent to cartilage canals. Given the initial distribution of label retaining cells after the 3 day chase, we speculate that these core chondrocytes derive from label-retaining progenitors in the perichondrium and were transported to the cartilaginous core of the metapterygium via cartilage canals (*Figure 9*). However, skate cartilage canals also contain red blood cells (*Figure 4h*), which suggests that these structures may serve to vascularise the adult cartilaginous endoskeleton. If this is the case, then such vasculature could serve to transport cartilage progenitor or mesenchymal stem cell-like cells, from niches elsewhere in body, to the core cartilage of the metapterygium.

Cartilage canals that extend from the perichondrium into the core of endoskeletal elements have been shown to occur in the vertebrae, jaws and pectoral girdles of sharks and rays (*Hoenig and Walsh, 1982*; *Dean et al., 2010*). Similar canals have been described from the cartilage of embryonic tetrapods, though in tetrapods these are transient structures that function in mediating the replacement of cartilage by bone during endochondral ossification (*Blumer et al., 2004*; *Blumer et al., 2008*). Conversely, cartilage canals in cartilaginous fishes persist within the adult skeleton, have been described as containing blood vessels and lymph-like and other amorphous materials, as well as immature chondrocytes, and have been speculated to function in the nourishment and maintenance of cartilage in the adult endoskeleton (*Hoenig and Walsh, 1982*; *Dean et al., 2010*). The cartilage canals that we have described in the metapterygium of the skate closely resemble those described previously in other cartilaginous fishes, and findings from our label retention experiments are consistent with a function for these canals in transporting material (including pre-chondrocytes) to the core cartilage of the metapterygium. It therefore seems likely that cartilage canals do,

indeed, function in the maintenance, nourishment, and, most likely, interstitial growth of cartilage in the endoskeleton of cartilaginous fishes.

## Spontaneous repair of hyaline cartilage in the skate

We have found that the persistence of cartilage progenitor cells and chondrogenesis in the adult skate skeleton correlates with an ability to spontaneously repair injured cartilage – albeit with a tissue containing a significantly higher density of chondrocytes relative to normal adult cartilage. A comparative analysis of the mechanical properties of normal and repair cartilage remains to be conducted, though the ECM of skate repair cartilage is composed of type II collagen and appears to integrate seamlessly with adjacent tissue, and cells within repair cartilage appear indistinguishable from adjacent chondrocytes. While it is not currently possible to precisely trace the cell lineage of repair cartilage within our injury paradigm, the demonstrated chondrogenic potential of the adult skate perichondrium during normal growth, as well as our observation that mechanical perturbation of the surface of the metapterygium is sufficient to induce a large mass of ectopic cartilage beneath the perichondrium, points to the perichondrium as the most likely source of repair cartilage following injury.

Our findings stand in contrast with previous work in the dogfish (*Scyliorhinus spp.*), which determined that the cartilaginous skeleton of sharks could not heal following injury (*Ashhurst, 2004*). This was based on a design in which cartilaginous fin radials were bisected and monitored for repair over 26 weeks. It was observed that cut surfaces of dogfish fin rays were initially capped by a fibrous tissue, with subsequent appearance near the injury site of a disorganized, cartilage-like tissue that failed to integrate with existing ray cartilage or to unite the bisected elements. Importantly, the cartilage injury in that study (i.e. complete bisection) was severe, and may have posed an insurmountable barrier to repair, even for a tissue with local progenitors and repair potential (i.e. it is possible that the repair response that we report in skate requires some scaffold of normal cartilage as a foundation for repair). Additionally, fin rays are relatively small, and therefore may exhibit different growth properties relative to larger elements of the endoskeleton. Additional studies of repair potential across the range of skeletal elements and tissue types in cartilaginous fishes are needed to determine whether cartilage repair is a general feature of the skeleton, or a unique property of specific skeletal elements.

## Skate as a model for adult chondrogenesis and cartilage repair

Osteoarthritis (OA) is a debilitating deterioration of joint cartilage with symptoms ranging from stiffness and joint pain to complete immobility. OA can severely impact quality of life, and has an extremely high economic burden, and so there is great interest in identifying novel therapeutic strategies to promote joint cartilage repair. Joint cartilage repair still poses a substantial clinical challenge, owing to the avascular and aneural nature of articular cartilage, and therefore its limited capacity to initiate spontaneous repair. Recently, focus has shifted from surgical approaches (e.g. microfracture and autologous chondrocyte implantation – *Rodrigo et al., 1994*; *Brittberg et al., 1994*) to stem cell-based therapies for cartilage defects – namely the application of patient-derived mesenchymal stem cells (MSCs) or induced pluripotent stem cells (iPSCs) as sources of repair tissue for damaged cartilage (*Bernhard and Vunjak-Novakovic, 2016*; *Murphy et al., 2017*; *Harrell et al., 2019*). While these approaches hold great promise, some challenges nevertheless remain. Derivation of persistent cartilage from MSCs is challenging, as chondrogenically differentiated MSCs will often continue on a path toward hypertrophy and ultimately ossification (*Pelttari et al., 2006*; *Steinert et al., 2007*), while the use of incompletely differentiated iPSCs can result in heterogeneous repair tissue, and may come with a risk of teratoma formation (*Heng et al., 2004*; *Saito et al., 2015*).

The unique endoskeletal growth and repair properties of cartilaginous fishes may offer a powerful model to inform novel cell-based strategies for mammalian cartilage repair. It remains to be determined whether the adult skate perichondrium is a homogeneous cell population with equivalent chondrogenic potential throughout, or a heterogeneous tissue containing a specialized subpopulation of chondroprogenitors. However, the ability of this tissue to give rise to cartilage as a terminal product – rather than cartilage as an intermediate step toward endochondral bone – is significant and could be exploited to further our understanding of the molecular basis of stable and reliable

adult chondrogenesis in vitro and in vivo. Further characterization of the transcriptional fingerprint of cell types within the skate perichondrium, and the gene regulatory basis skate chondrocytes differentiation during growth and repair, could provide a rich source of information on how MSCs and/or iPSCs could be edited or manipulated to enhance their efficacy in mammalian articular cartilage repair.

## Materials and methods

### Animals, euthanasia and fixation

*Leucoraja erinacea* adults and embryos were maintained in large rectangular tanks or seatables, respectively, in flow-through natural seawater at a constant temperature of 15˚C at the Marine Resources Center of the Marine Biological Laboratory in Woods Hole, MA, U.S.A. Adult skates were fed on a diet of squid and capelin. Skate embryos were staged according to the lesser-spotted dogfish (*Scyliorhinus canicula*) staging table of *Ballard et al. (1993)* and the winter skate (*Leucoraja ocellata*) staging table of *Maxwell et al. (2008)*. Prior to fixation, all skate embryos and adults were euthanized with an overdose of ethyl 3-aminobenzoate methanesulfonate salt (MS-222 - Sigma) in seawater (1 g/L MS-222 buffered with 2 g/L sodium bicarbonate). Animals were kept in a euthanasia bath until the cessation of gill pumping and heartbeat, and for adults, decapitation was used as a secondary method of euthanasia.

All skate embryos were fixed overnight in 4% paraformaldehyde (Electron Microscopy Science) in 1X phosphate buffered saline (PBS – ThermoFisher), rinsed 3 × 10 min in 1X PBS, dehydrated stepwise into 100% methanol, and stored in methanol at −20˚C prior to analysis. Pieces of dissected adult skate cartilage to be used for histochemical staining, immunofluorescence or EdU detection were fixed in 4% paraformaldehyde in filtered seawater for 48 hr, rinsed into filtered seawater containing ~0.5% paraformaldehyde and stored at 4˚C. Pieces of dissected adult skate cartilage to be used for mRNA in situ hybridization were fixed for 48 hr in 4% paraformaldehyde in 1X PBS, then rinsed 3 × 10 min in 1X PBS, dehydrated stepwise into 100% methanol, and stored in methanol at −20˚C.

All work involving skate embryos and adults was conducted in strict accordance with protocols approved by the Marine Biological Laboratory Institutional Animal Care and Use Committee.

### Paraffin histology and histochemical staining

For paraffin embedding of embryonic tissue, specimens were cleared 3 × 20 min in Histosol (National Diagnostics) at room temperature, transitioned through 2 × 30 min steps in 1:1 Histosol: molten paraffin in a standard wax oven at 60˚C, then left in molten paraffin (RA Lamb Wax – Fisher Scientific) at 60˚C overnight. The following day, specimens were moved through four changes of molten paraffin (each >1 hr) before positioning and embedding in a Peel-A-Way embedding mold (Sigma).

For paraffin embedding of adult skate tissue, samples were rinsed several times in water, and then demineralised in a 10%(w/v) solution of ethylenediaminetetraacetic acid (EDTA) in 0.1M Tris pH 7.2 for 20–25 days on a rocking platform at 4˚C. Upon completion of demineralisation, samples were washed several times with water, and then infiltrated with paraffin under vacuum in a tissue processor (with stepwise dehydration into 100% ethanol, 3 × 1 hr washes in xylene and 4 × 45 min washes in molten paraffin) before embedding in a Peel-A-Way embedding mold.

All blocks were left to set overnight at room temperature before sectioning at 8 μm on a Leica RM2125 rotary microtome. Sections were mounted on Superfrost plus slides (VWR) and stained with a modified Masson's trichrome stain, according to the protocol of *Witten and Hall (2003)*. All histochemical staining was carried out at least in triplicate (i.e. on three separate stage-matched individuals).

### Immunofluorescence

Slides to be used for immunofluorescence were dewaxed in histosol and rehydrated through a descending ethanol series into 1X PBS + 0.1% Triton X-100 (PBT). For enzymatic antigen retrieval, slides were incubated in 147 u/mL hyaluronidase (Sigma) in PBS pH 6.7 for 1 hr at 37˚C followed by 0.1% (w/v) pepsin (Sigma) in 0.01N HCl for 30 min at 37˚C for anti-Col2a1 or 0.1% (w/v) pepsin in

0.5M acetic acid for 2 hr at 37˚C for anti-Col1a1, according to the protocol of *Egerbacher et al. (2006)*. Slides were then rinsed 3 × 10 min in PBT, blocked for 30 min in 10% sheep serum and incubated in primary antibody (under a parafilm coverslip) in a humidified chamber overnight at 4˚C. The following day, slides were rinsed 3 × 5 min in 1X PBT, and then incubated in secondary antibody (under a parafilm coverslip) in a humidified chamber overnight at 4˚C. Slides were then rinsed 3 × 10 min in PBT and coverslipped with Fluoromount G containing DAPI (Southern Biotech). Primary and secondary antibodies were diluted in 10% sheep serum in PBT to the following concentrations: anti-COL2A1 (II.II6B3, Developmental Studies Hybridoma Bank, University of Iowa; 1:20), anti-COL1A1 (LF-68, Kerafast; 1:100), AF568-conjugated goat-anti-rabbit IgG (A11011, Invitrogen; 1:500) and AF488-conjugated goat-anti-mouse IgG (A11001, Invitrogen; 1:500). All immunostaining was carried out in triplicate (on three separate stage-matched individuals), and negative controls were conducted by following the same staining protocol but in the absence of primary antibody.

## EdU incorporation experiments

5-ethynyl-2'-deoxyuridine (EdU – ThermoFisher Scientific) retention experiments were conducted to label the nuclei of cell that have undergone S-phase DNA synthesis and their progeny (*Salic and Mitchison, 2008*). For skate hatchling EdU retention experiments, hatchlings were anaesthetized in MS-222 in seawater (150 mg/L MS-222 buffered with 300 mg/L sodium bicarbonate) and given a single intraperitoneal (IP) microinjection of 2 µL of a 5 mM EdU solution in 1X PBS with a Picospritzer pressure injector. Animals were then recovered in seawater and reared in a flow-through seatable for 1, 5, 10 or 40 days, at which point animals were euthanized and fixed as described above.

For adult skate EdU pulse-chase experiments, eight adults weighing between 750–850 g were given three 5 mL IP injections of 22 mM EdU in 1X PBS (a dose of 27.7 mg EdU/injection), with 48 hr between each injection. Animals were anaesthetized in MS-222 in aerated seawater (150 mg/L MS-222 buffered with 300 mg/L sodium bicarbonate) prior to injection, and after injection, animals were recovered in aerated seawater before being returned to their tank. Two animals were euthanized, dissected and fixed as described above 3 days, 1 month, 2 months and 5.5 months after the final EdU injection.

EdU detection was carried out on 8 µm paraffin sections using the Click-iT EdU Cell Proliferation Kit (ThermoFisher Scientific) according to the manufacturer's instructions. After detection, slides were coverslipped with Fluoromount G containing DAPI, imaged, then de-coverslipped in water and stained with modified Masson's trichrome. Given the large size of the metapterygium and the relative sparsity of certain label-retaining cell types (e.g. chondrocytes in adult sections), cell counts were performed across five successive sections through the region of the metapterygium indicated in *Figure 1* for adult skates, and three successive sections through the equivalent region of the metapterygium in hatchlings. For counts of EdU+ perichondral cells in skate hatchlings, the sum total of EdU+ cells for each individual were plotted against chace time in MS Excel, and a polynomial trend line was added.

## mRNA in situ hybridization and phylogenetic analysis

For paraffin embedding and sectioning of adult tissues for mRNA in situ hybridization, tissues were rehydrated into diethyl pyrocarbonate (DEPC)-treated water and demineralised for 24 hr in Morse Solution (5 g sodium citrate dihydrate, 12.5 mL formic acid and 37.5 mL DEPC water). Demineralised tissues were then dehydrated stepwise into 100% ethanol before infiltration, paraffin embedding and sectioning as described above.

Chromogenic mRNA in situ hybridization on paraffin sections for *Leucoraja erinacea Col2a1* (GenBank MT254563) and *Agc* (GenBank MT254564) was carried out according to the protocol of *O'Neill et al. (2007)*, with modifications according to *Gillis et al. (2012)*. mRNA in situ hybridization by chain reaction (HCR) was carried out according to the protocol of *Choi et al. (2018)* with the following modifications: slides were pre-hybridized for 30 min at 37˚C; for the hybridization step, 0.8 µL of each 1 µM probe stock was used per 100 µL of hybridization buffer; and hairpins were used at 4 µL of hairpin stock per 100 µL of amplification buffer. Probe sets for *Leucoraja erinacea Col2a1* (GenBank MT254563), *Sox9* (GenBank MT254560), *Sox5* (GenBank MT254561) and *Sox6* (GenBank MT254562) and hairpins were purchased from Molecular Instruments. Molecular Instrument probe

lot numbers are as follows: *Col2a1*(PRB574), *Sox9* (PRB571), *Sox5* (PRB572) and *Sox6* (PRB573). All mRNA in situ hybridization slides were coverslipped with Fluoromount G containing DAPI.

Orthology of the *Leucoraja erinacea Col2a1*, *Agc*, *Sox9* and *Sox6* sequences used for probe design was confirmed by phylogenetic analysis. Full or partial coding sequences were translated using ORFfinder (NCBI), and multiple sequence alignments with aggrecan, clade A collagen, SoxD and SoxE protein family members were constructed using Clustal Omega (*Sievers and Higgins, 2018*). The alignments were trimmed with TrimAl (*Capella-Gutiérrez et al., 2009*) and subsequently used to infer evolutionary relationships with maximum likelihood method in IQ-TREE v1.6.12 (*Nguyen et al., 2015*). ModelFinder (*Kalyaanamoorthy et al., 2017*) implemented in IQ-TREE was used to find the best-fit substitution model based on the Bayesian information criterion (BIC) for each protein alignment. The branch supports for ML analyses were obtained using the ultrafast boot-strap (UBS) (*Minh et al., 2013*) with 1000 replicates. Phylogenetic trees (*Figure 3—figure supplement 1–4*) were prepared using iTOL v5 (*Letunic and Bork, 2019*) and bootstrap values below or equal to 75% are shown.

The following amino acid sequences were retrieved from GenBank for inclusion in our phylogenetic analyses:

Mouse (*Mus musculus*) Sox5, XP_006506994; human (*Homo sapiens*) Sox5, NP_008871; chick (*Gallus gallus*) Sox5, XP_015145677; rat (*Rattus norvegicus*) Sox5, XP_006237666; zebrafish (*Danio rerio*) Sox5, XP_021330769; cow (*Bos taurus*) Sox5, XP_005207008; frog (*Xenopus tropicalis*) Sox5, XP_031753364; dog (*Canis lupus familiaris*) Sox5, XP_022267096; elephant fish (*Callorhinchus milii*) Sox5, XP_007895487; mouse (*Mus musculus*) Sox6, XP_006507558; human (*Homo sapiens*) Sox6, NP_001354802; rat (*Rattus norvegicus*) Sox6, XP_006230149; zebrafish (*Danio rerio*) Sox6, NP_001116481; chick (*Gallus gallus*) Sox6, XP_025006442; frog (*Xenopus tropicalis*) Sox6, XP_031755685; cow (*Bos taurus*) Sox6, XP_024831142; dog (*Canis lupus familiaris*) Sox6, XP_022263574; elephant fish (*Callorhinchus milii*) Sox6, XP_007885710; mouse (*Mus musculus*) Sox8, NP_035577; human (*Homo sapiens*) Sox8, NP_055402; rat (*Rattus norvegicus*) Sox8, NP_001100459; chick (*Gallus gallus*) Sox8, NP_990062; cow (*Bos taurus*) Sox8, XP_002698019; frog (*Xenopus tropicalis*) Sox8, XP_002932315; dog (*Canis lupus familiaris*) Sox8, XP_022275986; horse (*Equus caballus*) Sox8, XP_005599176; elephant fish (*Callorhinchus milii*) Sox8, XP_007901694; Mouse (*Mus musculus*) Sox9, NP_035578; human (*Homo sapiens*) Sox9, NP_000337; chick (*Gallus gallus*) Sox9, NP_989612; rat (*Rattus norvegicus*) Sox9, NP_536328; zebrafish (*Danio rerio*) Sox9, NP_571718; cow (*Bos taurus*) Sox9, XP_024836864; frog (*Xenopus tropicalis*) Sox9, NP_001016853; dog (*Canis lupus familiaris*) Sox9, NP_001002978; horse (*Equus caballus*) Sox9, XP_023507898; mouse (*Mus musculus*) Sox10, NP_035567; human (*Homo sapiens*) Sox10, NP_008872; chick (*Gallus gallus*) Sox9, XP_015139949; rat (*Rattus norvegicus*) Sox10, NP_062066; zebrafish (*Danio rerio*) Sox10, NP_571950; cow (*Bos taurus*) Sox10, NP_001180176; frog (*Xenopus tropicalis*) Sox10, NP_001093691; dog (*Canis lupus familiaris*) Sox10, XP_538379; horse (*Equus caballus*) Sox10, XP_023487097; fruit fly (*Drosophila melanogaster*) Sox100B, NP_651839; fruit fly (*Drosophila melanogaster*) Sox102F, NP_001014695; mouse (*Mus musculus domesticus*) aggrecan, AAC37670; human (*Homo sapiens*) aggrecan, AAH36445; cow (*Bos taurus*) aggrecan, AAP44494; rat (*Rattus norvegicus*) aggrecan, AAA21000; zebrafish (*Danio rerio*) aggrecan, XP_021326217; elephant fish (*Callorhinchus milii*) aggrecan, XP_007906559; frog (*Xenopus tropicalis*) aggrecan, XP_017948155; horse (*Equus caballus*) aggrecan, XP_005602856; mouse (*Mus musculus domesticus*) Col2α1, NP_112440; rat (*Rattus norvegicus*) Col2α1, XP_006242370; human (*Homo sapiens*) Col2α1, NP_001835; cow (*Bos taurus*) Col2α1, NP_001001135; chick (*Gallus gallus*) Col2α1, XP_025001042; zebrafish (*Danio rerio*) Col2α1a, NM_131292; dog (*Canis lupus familiaris*) Col2α1, NP_001006952; horse Col2α1 (*Equus caballus*), XP_005611139; frog (*Xenopus tropicalis*) Col2α1, NP_989220; elephant fish (*Callorhinchus milii*) Col2α1, XP_007908719; sea lamprey (*Petromyzon marinus*) Col2α1a, ABB53637; sea lamprey (*Petromyzon marinus*) Col2α1b, ABB53638; mouse (*Mus musculus domesticus*) Col1α1, CAI25880; human (*Homo sapiens*) Col1α1, BAD92834; zebrafish (*Danio rerio*) Col1α1, AAH63249; dog (*Canis lupus familiaris*) Col1α1, NP_001003090; cow (*Bos taurus*) Col1α1, AAI05185; frog (*Lithobates catesbeianus*) Col1α1, BAA29028; mouse (*Mus musculus domesticus*) Col1α2, NP_031769; human (*Homo sapiens*) Col1α2, AAH42586; chick (*Gallus gallus*) Col1α2, XP_418665; dog (*Canis lupus familiaris*) Col1α2, NP_001003187; frog (*Xenopus laevis*) Col1α2, AAH49287; zebrafish (*Danio rerio*) Col1α1, NP_892013; human (*Homo sapiens*) Col3α1, AAL13167; dog (*Canis lupus familiaris*) Col3α1, XP_851009; frog (*Xenopus laevis*) Col3α1, AAH60753; cow (*Bos taurus*) Col3α1, NP_001070299;

mouse (*Mus musculus* domesticus) Col5α2, NP_031763; human (*Homo sapiens*) Col5α2, NP_000384; chick (*Gallus gallus*) Col5α2, XP_015144688; dog (*Canis lupus familiaris*) Col5α2, XP_535998; cow (*Bos taurus*) Col5α2, XP_581318; rat (*Rattus norvegicus*) Col5α2, NP_445940; sea urchin (*Strongylocentrotus purpuratus*) ColP2α, NP_999675; tunicate (*Ciona intestinalis*) fCol1, XP_026690723; acorn worm (*Saccoglossus kowalevskii*) fibrillar collagen, ABB83364.

The sea lamprey aggrecan-like protein sequence (ENSPMAP00000001826) was retrieved from *P. marinus* Ensembl genome assembly (Pmarinus_7.0) based on BLAST searches for sequence conservation. Similarly, tunicate clade A fibrillar collagen gene (ci0100150759) was retrieved from JGI *C. intestinalis* genome assembly (C.intestinalis V2.0).

The *Leucoraja erinacea Sox5* sequence corresponds with the 3' untranslated region (UTR) of the *Sox5* transcript, and so orthology could not be confirmed as described above. However, this sequence showed significant homology with the 3' UTR of predicted *Sox5* transcripts from other chondrichthyan species (*Rhincodon typus*, *Amblyraja radiata* and *Callorhinchus milii*) by BLAST.

## Cartilage injury experiments

For cartilage injury experiments, adult skates ranging in weight from 500 to 750 g were anaesthetized in MS-222 in aerated seawater (150 mg/L MS-222 buffered with 300 mg/L sodium bicarbonate) until the animals failed to respond to noxious stimulus (e.g. a pinch with forceps). Anaesthetized animals were given a pre-operative analgesic (0.25 mg butorphanol by intramuscular injection) and then moved from the anesthesia bath to an operating table, where their gills were perfused with anesthetic seawater for the duration of the procedure (~5 min). A small (~2 cm) surgical incision was made through the dorsal surface of the fin,~3 cm from the base of the metapterygium, and a wedge of cartilage was removed from the metapterygium using a 4 mm biopsy punch. Following biopsy, the incision was sutured, animals were given a postoperative dose of antibiotic (15 mg ceftazidime by intramuscular injection) and then recovered in aerated seawater until fully awake before returning to their holding tank. No animals died as a result of the procedure. Two animals were collected one-week post-operation, and then at monthly intervals for the following year. All animals were euthanized, dissected, fixed and processed for histological analysis as described above.

## MicroCT scanning

Tissue samples from adult skates collected one-week post-biopsy were imaged by X-ray microtomography (microCT) at the Cambridge Biotomography Centre (Department of Zoology, University of Cambridge). Samples were scanned using a Nikon XTH225 ST scanner, at 100kV and a 120microamps beam current.

## Microscopy and image analysis

All microscopy was performed with a Zeiss Axioscope A1 and Zen software. All images were processed and plates prepared using Adobe Photoshop CC and Adobe Illustrator CC.

## Acknowledgements

The authors acknowledge Dr. Kate Rawlinson, Prof. Brian Hall, Dr. Kate Criswell, Dr. Victoria Sleight, Christine Hirschberger and Jenaid Rees for a collective many years of helpful discussion around the topic of cartilage development and repair, Janice Simmons, Dan Calzarette, Scott Bennett, David Remsen and the staff of the Marine Biological Laboratory Marine Resources Center for expert assistance with animal maintenance and care, and Helen Skelton (Dept. of Pathology, University of Cambridge) and Debbie Sabin (Dept. of Veterinary Medicine, University of Cambridge) for assistance with adult skate tissue processing. This work was funded by the Wellcome Trust (PhD studentship 102175/Z/13/Z to AM), the Royal Society (University Research Fellowships UF130182 and URF/R/191007 and Research Fellows Enhancement Award RGF\EA\180087 to JAG), the Isaac Newton Trust (award 14.23z to JAG) and by a research grant from the Fisheries Society of the British Isles (to JAG).

## Additional information

### Funding

| Funder | Grant reference number | Author |
|---|---|---|
| Wellcome | 102175/Z/13/Z | Aleksandra Marconi |
| Royal Society | UF130182 | J Andrew Gillis |
| Royal Society | URF/R/191007 | J Andrew Gillis |
| Royal Society | RGF\EA\180087 | J Andrew Gillis |
| Isaac Newton Trust | 14.23z | J Andrew Gillis |
| Fisheries Society of the British Isles | Research Grant | J Andrew Gillis |

The funders had no role in study design, data collection and interpretation, or the decision to submit the work for publication.

### Author contributions

Aleksandra Marconi, Investigation; Amy Hancock-Ronemus, Investigation, Methodology; J Andrew Gillis, Conceptualization, Supervision, Funding acquisition, Investigation, Visualization, Methodology, Project administration

### Author ORCIDs

J Andrew Gillis (iD) https://orcid.org/0000-0003-2062-3777

### Ethics

Animal experimentation: This study was performed in strict accordance with the recommendations in the Guide for the Care and Use of Laboratory Animals of the National Institutes of Health. All of the animals were handled according to approved institutional animal care and use committee (IACUC) protocols (#14-54 and 17-24) of the Marine Biological Laboratory. All surgery was performed under tricaine anesthesia, and every effort was made to minimize suffering.

### Decision letter and Author response

Decision letter https://doi.org/10.7554/eLife.53414.sa1
Author response https://doi.org/10.7554/eLife.53414.sa2

## Additional files

### Supplementary files

• Transparent reporting form

### Data availability

All cDNA sequences used to generate mRNA in situ hybridisation probes are available at NCBI: *Leucoraja erinacea Col2a1* (MT254563), *L. erinacea Agc* (MT254564), *L. erinacea Sox9* (MT254560), *L. erinacea Sox5* (MT254561) and *L. erinacea* (MT254562). All histological, immunofluorescent and in situ hybridisation data are presented in the figures.

The following datasets were generated:

| Author(s) | Year | Dataset title | Dataset URL | Database and Identifier |
|---|---|---|---|---|
| Marconi A, Hancock-Ronemus A, Gillis JA | 2020 | BankIt2325308 Seq1 | https://www.ncbi.nlm.nih.gov/nuccore/MT254560 | NCBI GenBank, MT254560 |
| Marconi A, Hancock-Ronemus A, | 2020 | BankIt2325308 Seq2 | https://www.ncbi.nlm.nih.gov/nuccore/ | NCBI GenBank, MT254561 |

| | | | | |
|---|---|---|---|---|
| Gillis JA | | | MT254561 | |
| Marconi A, Han-cock-Ronemus A, Gillis JA | 2020 | BankIt2325308 Seq3 | https://www.ncbi.nlm.nih.gov/nuccore/MT254562 | NCBI GenBank , MT254562 |
| Marconi A, Han-cock-Ronemus A, Gillis JA | 2020 | BankIt2325308 Seq4 | https://www.ncbi.nlm.nih.gov/nuccore/MT254563 | NCBI GenBank , MT254563 |
| Marconi A, Han-cock-Ronemus A, Gillis JA | 2020 | BankIt2325308 Seq5 | https://www.ncbi.nlm.nih.gov/nuccore/MT254564 | NCBI GenBank , MT254564 |

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
