## [Decision Letter]

**Acceptance summary:**

The studies on *L. erinacea* (the Little Skate) described in this manuscript report findings of great importance, including the conservation of cartilage formation and growth, from chondrichthyan fishes to mammals – an important connection for biomedical inference. Because these animals display constitutive growth, cartilage development is likely a perpetual event, providing a unique opportunity to study how cartilage can be repaired after damage. As such, this works highlights the need for models that can offer insights into how vertebrates can regenerate important skeletal tissues.

**Decision letter after peer review:**

Thank you for submitting your article "Adult chondrogenesis and spontaneous cartilage repair in the skate, *Leucoraja erinacea*" for consideration by *eLife*. Your article has been reviewed by two peer reviewers, and the evaluation has been overseen by Alejandro Sánchez Alvarado as the Reviewing Editor and Clifford Rosen as the Senior Editor. The following individual involved in review of your submission has agreed to reveal their identity: Melanie Debiais Thibaud (Reviewer #2).

The reviewers have discussed the reviews with one another and the Reviewing Editor has drafted this decision to help you prepare a revised submission.

Summary:

This manuscript by Marconi et al. lays down the foundation for a new research organism to study chondrogenesis and cartilage regeneration by exploiting the ability of *L. erinacea* (the Little Skate) to retain active chondrogenesis beyond embryogenesis. The manuscript reports findings of great importance, including the conservation of cartilage formation and growth, from chondrichthyan fishes to mammals – an important connection for biomedical inference. Because these animals display constitutive growth, cartilage development is likely a perpetual event, providing a unique opportunity to study how cartilage can be repaired after damage, highlighting the need for models that can offer insights into how vertebrates can regenerate important skeletal tissues.

Essential revisions:

In order for this manuscript to be considered further, the following points need to be addressed fully:

1) The authors need to provide phylogenetic evidence supporting either homology or orthology for the probes used to detect *Col2a1*, *Sox9*, *Sox5*, and *Sox6*. As presented now, all we know about these probes is that they were purchased from Molecular Instruments (subsection “mRNA in situ hybridization”). Sequences for the probes used, as well as an appropriate phylogenetic analysis, are requisite in order to ascertain both identity and specificity.

2) The authors speculate that "…these core chondrocytes derive from label-retaining progenitors in the perichondrium and were transported". While this is a reasonable assessment, insufficient data is provided to resolve this matter unambiguously. Therefore, the authors should either provide alternative explanations for the origin of the circulating cells, or better justify/support why they consider only the perichondrium as a source of these cells.

3) Along the same lines, the authors should modify their Abstract, in which they write: "perichondral progenitor cells that generate new cartilage during adult growth". This statement is not strictly demonstrated by the data presented.

4) Why did the authors not compare the retention of EdU in hatchling skates compared to adults? We understand the need to show this in the adult, but as this is a continuation of the 'developmental process' we should, therefore, have a comparative baseline of cartilage development and retention of the label. It seems there is a disconnect between the discussion of embryonic/hatchling tissue and then adult. This would permit disentangling the relationship between true development and adult growth (or prolonged development).

5) Similarly, and probably less important, there is no comparison of spontaneous damage repair in embryonic tissues. How does repair differ between embryos and adults? Would the overall mechanisms be similar? And therefore, is the repair during the growth/extended development observed in adults similar to repair during the development of the structures? If the authors have this data please consider its inclusion. Alternatively, the authors could provide some extended discussion of this omission instead.

6) Discussion. The connection between this work and studies in Osteoarthritis is intriguing, and absolutely worth a mention for the translational potential here; however, is the entire discussion on OA absolutely necessary? If the association between skate chondrogenesis and OA is not clearly presented at the start of this Discussion section, then the 'review' of OA in human patients loses the relevance of the skate affiliation. A clearly defined translational implication should be stated and this section should be kept brief to avoid the risk of overselling the implications of the work, which is in its infancy here and has no direct relationship to OA.

---

## [Author Response]

Essential revisions:In order for this manuscript to be considered further, the following points need to be addressed fully:1) The authors need to provide phylogenetic evidence supporting either homology or orthology for the probes used to detect Col2a1, Sox9, Sox5, and Sox6. As presented now, all we know about these probes is that they were purchased from Molecular Instruments (subsection “mRNA in situ hybridization”). Sequences for the probes used, as well as an appropriate phylogenetic analysis, are requisite in order to ascertain both identity and specificity.

We now include a complete phylogenetic analysis of our skate *Col2a1*, *Agc*, *Sox9* and *Sox6* sequences, with new text in the Materials and methods section describing the analysis (subsection “mRNA in situ hybridization and phylogenetic analysis”), and the resulting phylogenetic trees as new supplements to Figure 3 (Figure 3—figure supplements 1-4). These analyses fully support the orthology of our sequences. For *Sox5*, our sequence falls entirely within the 3’ UTR, so a phylogenetic analysis of coding sequence was not possible. However, we are nevertheless confident with the orthology of this sequence based on the very low BLAST scores (0 to 7e^-70^) for this sequence against other 3’ UTR sequences of predicted *Sox5* transcripts from other chondrichthyan genomes.

Molecular Instruments does not give us the sequences of the probes that they design for each target, as probes are designed using a proprietary algorithm. However, MI does test probe sequences against the skate genome to ensure that there is no off-target binding. Additionally, we have now included probe lot numbers in the Materials and methods section, so that others can purchase the exact probes that we used, should they wish to repeat this analysis.

Finally, we have now obtained and included GenBank accession numbers for all of the sequences used in our paper.

2) The authors speculate that "…these core chondrocytes derive from label-retaining progenitors in the perichondrium and were transported". While this is a reasonable assessment, insufficient data is provided to resolve this matter unambiguously. Therefore, the authors should either provide alternative explanations for the origin of the circulating cells, or better justify/support why they consider only the perichondrium as a source of these cells.

We agree with this comment, and we have adjusted the text accordingly. We speculate that the new core chondrocytes arise from perichondral progenitors, given that these progenitors clearly give rise to peripheral chondrocytes, and that the cartilage canals containing these pre-chondrocytes originate in the perichondrium. However, we also note in the text that skate cartilage canals contain red blood cells – and so, if skate cartilage is vascularized, it is absolutely possible that cartilage progenitors could be transported to the core of the metapterygium from elsewhere in the body. We have therefore added new text to the Discussion (subsection “Adult chondroprogenitor cells contribute to post-embryonic growth of cartilage in the skate”) to clarify this point.

3) Along the same lines, the authors should modify their Abstract, in which they write: "perichondral progenitor cells that generate new cartilage during adult growth". This statement is not strictly demonstrated by the data presented.

As with the previous point, we agree that we cannot say conclusively that perichondral progenitor cells give rise to the new chondrocytes that form in the core of the metapterygium adjacent to the blind end of cartilage canals. However, the vast majority of new (i.e. label-retaining) chondrocytes that we observe at 5.5 months are found in the periphery (i.e. in the hyaline cartilage of the intertesseral joint region). These cells sit adjacent to the perichondrium (and not in close proximity to cartilage canals or any other vasculature), and so it seems to us that the label-retaining cells of the perichondrium that we observe at the 1-day and 1-month chase time points are the only plausible source of these new chondrocytes. In our opinion, this is the most significant finding of the paper, and we believe that it is quite well supported by our data – so we would therefore like to keep this line in the Abstract (if the reviewing editor and reviewers agree?).

4) Why did the authors not compare the retention of EdU in hatchling skates compared to adults? We understand the need to show this in the adult, but as this is a continuation of the 'developmental process' we should, therefore, have a comparative baseline of cartilage development and retention of the label. It seems there is a disconnect between the discussion of embryonic/hatchling tissue and then adult. This would permit disentangling the relationship between true development and adult growth (or prolonged development).

We thank the reviewer for this great suggestion. We actually did perform a label retention experiment with skate hatchlings during one of our past field seasons in Woods Hole, and we intended to use this experiment as part of a different study. However, we agree that it could be very informative here, and so we have carried out a series of new analyses (and prepared a new Figure 4), which we present in the Results subsection “Proliferation of chondrocytes and putative perichondral progenitor cells in the metapterygium of skate hatchlings”. We are really excited about these findings, as we think that they add a very nice dataset to the study.

Briefly: we conducted EdU pulse chase experiments by giving skate hatchlings an intraperitoneal microinjection of ~100µg EdU, and then harvesting at 1-, 5-, 10- and 40-days post injection. At 1dpi, we find EdU retention both within the chondrocytes of the metapterygium, and within the perichondrium. We quantified label-retaining cells within the perichondrium and found a striking increase in the number of EdU+ perichondral cells from 1 to 10 dpi, but then a decrease at 40dpi. We take this as an indication that perichondral cells are proliferating as the metapterygium grows, resulting in an increase in the number of EdU+ cells, until the EdU is eventually diluted beyond the limits of detectability (accounting for the decrease in EdU+ cells at 40dpi). We also observe clusters of EdU+ chondrocytes immediately beneath EdU+ perichondral cells at 10 and 40dpi, and we propose that these new chondrocytes are therefore likely of perichondral origin. Finally, we repeated our multiplexed ISH experiment with sections of metapterygium from skate hatchlings, and (as in adult skates) we observe flattened perichondral cells at the perichondral-cartilage interface, which co-express *Sox9*, *Sox5* and *Sox6* (but not *Col2a1*). We speculate that these are the same perichondral progenitor cells that we have identified through label retention and HCR in the adult skate metapterygium. So, overall, it appears as though hatchling skate cartilage grows both through continued proliferation of differentiated chondrocytes, and through recruitment of new chondrocytes from perichondral progenitor cells – but that by adulthood, new chondrocytes arise only from perichondral progenitors.

5) Similarly, and probably less important, there is no comparison of spontaneous damage repair in embryonic tissues. How does repair differ between embryos and adults? Would the overall mechanisms be similar? And therefore, is the repair during the growth/extended development observed in adults similar to repair during the development of the structures? If the authors have this data please consider its inclusion. Alternatively, the authors could provide some extended discussion of this omission instead.

I am afraid we do not have these data and cannot generate them in a timely manner (as it stands now, our summer 2020 Woods Hole field season has been cancelled due to the ongoing Covid-19 pandemic). The overall aim of our paper was to investigate whether/how cartilage in skates undergoes continued post-embryonic growth and repair, as the persistence of a cartilaginous endoskeleton throughout life is really what sets chondrichthyans apart from other jawed vertebrates. While it would be interesting to look at mechanisms of skeletal repair in embryos, we would not be surprised if such repair occurred (as even many bony vertebrate taxa can repair damage to embryonic or early juvenile cartilaginous structures). In our opinion, the most exciting products of this study are the demonstrations of chondrogenesis and repair in adult animals. However, we hope that our inclusion of new data on cell proliferation and gene expression within the hatchling metapterygium will go some way toward addressing the reviewer’s questions about mechanisms of cartilage growth in younger animals (though we fully acknowledge that it doesn’t shed light on mechanisms of repair at earlier life stages).

6) Discussion. The connection between this work and studies in Osteoarthritis is intriguing, and absolutely worth a mention for the translational potential here; however, is the entire discussion on OA absolutely necessary? If the association between skate chondrogenesis and OA is not clearly presented at the start of this Discussion section, then the 'review' of OA in human patients loses the relevance of the skate affiliation. A clearly defined translational implication should be stated and this section should be kept brief to avoid the risk of overselling the implications of the work, which is in its infancy here and has no direct relationship to OA.

We appreciate the reviewer’s point here, and we have cut this section down by ~1/3. We now have two paragraphs in this subsection of the Discussion: one outlining current challenge around the treatment of OA, and one about how novel animals models of adult chondrogenesis and cartilage repair could shed light on new therapeutic strategies. We have also updated the reference list to remove some of the references cited in the original version of this subsection. We feel that this streamlined subsection is much better than that original version, so we thank the reviewer for picking up on this.